# EPSP Synthase-Depleted Cells Are Aromatic Amino Acid Auxotrophs in *Mycobacterium smegmatis*

Mario Alejandro Duque-Villegas,[a,b] Bruno Lopes Abbadi,[a] Paulo Ricardo Romero,[a] Letícia Beatriz Matter,[a] Luiza Galina,[a,c] Pedro Ferrari Dalberto,[a,b] Valnês da Silva Rodrigues-Junior,[a] Rodrigo Gay Ducati,[a,d] Candida Deves Roth,[a] Raoní Scheibler Rambo,[a] Eduardo Vieira de Souza,[a,b] Marcia Alberton Perello,[a] Héctor Ricardo Morbidoni,[e] Pablo Machado,[a,b] Luiz Augusto Basso,[a,b,c] Cristiano Valim Bizarro[a,b]

aInstituto Nacional de Ciência e Tecnologia em Tuberculose (INCT-TB), Centro de Pesquisas em Biologia Molecular e Funcional (CPBMF), Pontifícia Universidade Católica do Rio Grande do Sul (PUCRS), Partenon, Porto Alegre, Brazil
bPrograma de Pós-Graduação em Biologia Celular e Molecular, PUCRS, Partenon, Porto Alegre, Brazil
cPrograma de Pós-Graduação em Medicina e Ciências da Saúde, PUCRS, Partenon, Porto Alegre, Brazil
dPrograma de Pós-Graduação em Biotecnologia, Universidade Do Vale Do Taquari - Univates, Universitário, Lajeado, Brazil
eLaboratorio de Microbiologia Molecular, Facultad de Ciencias Medicas, Universidad Nacional de Rosario, Rosario, Argentina

**ABSTRACT** The epidemiological importance of mycobacterial species is indisputable, and the necessity to find new molecules that can inhibit their growth is urgent. The shikimate pathway, required for the synthesis of important bacterial metabolites, represents a set of targets for inhibitors of *Mycobacterium tuberculosis* growth. The *aroA*-encoded 5-enolpyruvylshikimate-3-phosphate synthase (EPSPS) enzyme catalyzes the sixth step of the shikimate pathway. In this study, we combined gene disruption, gene knockdown, point mutations (D61W, R134A, E321N), and kinetic analysis to evaluate *aroA* gene essentiality and vulnerability of its protein product, EPSPS, from *Mycolicibacterium* (*Mycobacterium*) *smegmatis* (*Ms*EPSPS). We demonstrate that *aroA*-deficient cells are auxotrophic for aromatic amino acids (AroAAs) and that the growth impairment observed for *aroA*-knockdown cells grown on defined medium can be rescued by AroAA supplementation. We also evaluated the essentiality of selected *Ms*EPSPS residues in bacterial cells grown without AroAA supplementation. We found that the catalytic residues R134 and E321 are essential, while D61, presumably important for protein dynamics and suggested to have an indirect role in catalysis, is not essential under the growth conditions evaluated. We have also determined the catalytic efficiencies ($K_{cat}/K_m$) of recombinant wild-type (WT) and mutated versions of *Ms*EPSPS (D61W, R134A, E321N). Our results suggest that drug development efforts toward EPSPS inhibition may be ineffective if bacilli have access to external sources of AroAAs in the context of infection, which should be evaluated further. In the absence of AroAA supplementation, *aroA* from *M. smegmatis* is essential, its essentiality is dependent on *Ms*EPSPS activity, and *Ms*EPSPS is vulnerable.

**IMPORTANCE** We found that cells from *Mycobacterium smegmatis*, a model organism safer and easier to study than the disease-causing mycobacterial species, when depleted of an enzyme from the shikimate pathway, are auxotrophic for the three aromatic amino acids (AroAAs) that serve as building blocks of cellular proteins: L-tryptophan, L-phenylalanine, and L-tyrosine. That supplementation with only AroAAs is sufficient to rescue viable cells with the shikimate pathway inactivated was unexpected, since this pathway produces an end product, chorismate, that is the starting compound of essential pathways other than the ones that produce AroAAs. The depleted enzyme, the 5-enolpyruvylshikimate-3-phosphate synthase (EPSPS), catalyzes the sixth step of shikimate pathway. Depletion of this enzyme inside cells was performed by disrupting or silencing the EPSPS-encoding *aroA* gene. Finally, we evaluated the essentiality of specific residues

Address correspondence to Cristiano Valim Bizarro, cristiano.bizarro@pucrs.br.

The authors declare no conflicts of interest.

from EPSPS that are important for its catalytic activity, determined with experiments of enzyme kinetics using recombinant EPSPS mutants.

**KEYWORDS** gene silencing, chorismate, essentiality, vulnerability, CRISPRi, molecular genetics

Human tuberculosis (TB) is an important infectious disease that continues to be a public health threat worldwide. Despite global efforts to lower TB burden, which resulted in a 9% and 14% cumulative decline in incidence and mortality, respectively, between 2015 and 2019, the End TB Strategy milestones for 2025 are far from being reached (1). According to the 2020 World Health Organization (WHO) TB report, around 10 million people developed the disease and 1.2 million HIV-negative individuals died from TB in 2019 (1). In humans, the acid-fast *Mycobacterium tuberculosis* bacillus is the main causative agent of pulmonary TB, a fatal condition without treatment (1, 2). Although TB is considered a curable disease, with a success rate of approximately 85% for drug-susceptible strains, the worldwide spread of multidrug-resistant and rifampicin-resistant TB (MDR/RR-TB) poses a challenge to the current first-line treatment available and recommended by the WHO (1). Resistant cases of TB have been documented since the very initial use of streptomycin as the first anti-TB monotherapy in 1943 (3, 4). Therapies for MDR/RR-TB are complex, more expensive, more prolonged, and more toxic compared to those for drug-susceptible TB, and it is estimated that only 56% of MDR-TB cases reach a cure (1, 5).

The origin of resistant strains has been related to the acquisition of multiple molecular mechanisms that allow *M. tuberculosis* to evade the action of anti-TB drugs, mostly by mutations on drug targets (6). Therefore, the development of new anti-TB drugs relying on new mechanisms of action is needed. As the number of molecular targets for current bacterial agents is limited (7), there is a growing interest in finding and validating new molecular targets for drug development. Apart from being essential, a drug target should be vulnerable, which means that the incomplete inhibition of its activity should be sufficient to produce a lethal phenotype (8). Some genes/proteins were found to be essential but not vulnerable, prompting the need to use molecular genetic tools for studying the vulnerability as part of the target validation process (9, 10).

The shikimate pathway is considered an attractive target for developing new rational-based antimicrobial agents (11). It is essential for bacterial growth but absent in most animals, including mammals, favoring the development of selective inhibitors for pathogenic bacteria (12, 13). This pathway is composed of seven different enzymatic steps, leading to the production of chorismate, which is a precursor of naphthoquinones, menaquinones, and mycobactins, as well as folates, ubiquinones, tryptophan, tyrosine, and phenylalanine (12, 14, 15).

The 5-enolpyruvylshikimate-3-phosphate synthase (EPSPS; EC 2.5.1.19) is the sixth enzyme of the shikimate pathway. EPSPS is encoded by the *aroA* gene and catalyzes the transfer of the carboxyvinyl portion of phosphoenolpyruvate (PEP) to the carbon-5 hydroxyl group of shikimate-3-phosphate (S3P) to form enolpyruvyl shikimate-3-phosphate (EPSP) (16). The shikimate pathway was deemed essential in *M. tuberculosis* (14), but *aroA*-deficient cells were found to be auxotrophic for the aromatic amino acids L-tryptophan, L-phenylalanine, and L-tyrosine (AroAAs) in other bacterial species (17–19). Moreover, there is a lack of studies evaluating EPSPS vulnerability, which is a fundamental step in target validation for drug discovery purposes.

In this study, we show that the *aroA* gene from *Mycobacterium smegmatis*, when grown in a defined medium, is essential and its protein product (*Ms*EPSPS) is vulnerable only in the absence of AroAA supplementation. The *aroA* gene was found to be essential for *M. smegmatis* growth on LB or 7H10 medium but not on the latter supplemented with AroAAs. Furthermore, using a CRISPRi system (20), we rescued the growth impairment observed in *aroA*-knockdown strains by supplementing the growth medium with AroAAs, indicating that *Ms*EPSPS is a vulnerable target only in the absence of AroAA supplementation.

We also evaluated the essentiality of selected amino acid residues from *Ms*EPSPS. We found that the catalytic site residues R134 and E321 are essential, while D61 was found to be a nonessential residue. The latter corresponds to a residue suggested to play an indirect catalytic role in *Escherichia coli* EPSPS (21) and to be important for dynamical features of the *Mt*EPSPS ortholog from *M. tuberculosis* (22). Furthermore, we found that the D61W mutation does not affect the bacilli growth curve. To compare the phenotypic impact of these mutations with enzyme activity, wild-type (WT) and mutated versions of recombinant *Ms*EPSPS (D61W, R134A, E321N) were produced and purified and their kinetic activities were characterized and compared. We found that R134A, E321N, and the nonessential D61W mutation have pronounced effects on *Ms*EPSPS catalytic efficiency ($k_{cat}/K_m$). The absence of any impact on growth, despite the extensive reduction in catalytic efficiency observed *in vitro*, indicates that D61W mutation might have a milder impact on enzyme activity within cells. Overall, our mutational studies suggest that, under growth conditions where *aroA* is essential (absence of adequate amounts of AroAAs), its essentiality is dependent on *Ms*EPSPS activity.

## RESULTS

***In vitro* essentiality of *aroA* gene from *M. smegmatis*.** We performed gene disruption by allelic exchange mutagenesis to evaluate the essentiality of the *aroA* gene from *M. smegmatis* under three different growth conditions: in LB agar (rich medium), in 7H10 agar (defined medium), and in 7H10 agar supplemented with L-tryptophan, L-phenylalanine, and L-tyrosine (AroAAs) (Fig. 1A). We obtained viable mutants only in the third condition, indicating that this gene is nonessential when mycobacterial cells are supplied with adequate amounts of AroAAs. Only few colonies were recovered from pPR27::KO_*aroA* transformants of the pNIP40::Ø-integrated strain after selection for double crossover (DCO) events in either LB or 7H10 agar without supplementation. All of them were revealed to be yellow (catechol positive), indicating that they have escaped the selection steps and are not DCO mutants. In contrast, all the colonies obtained for pPR27::KO_*aroA* transformants from both strains (WT merodiploid and pNIP40::Ø-integrated strains) grown under 7H10 medium supplemented with AroAAs are DCO mutants (white colonies after catechol testing). This was further confirmed by PCR (Fig. 1B).

***aroA* knockdown with CRISPRi in *M. smegmatis*.** We assessed the vulnerability of *aroA* from *M. smegmatis* by using CRISPRi in different growth contexts. Using an in-house script written in Python, we found 12 targets in the nontemplate (NT) strand of the *aroA*-coding sequence. Three distinct sequences next to functional protospacer-adjacent motifs (PAMs; 5′-NAGCAT-3′, 5′-NNAGGAT-3′, and 5′-NNAGCAG-3′) and located at the first half of the gene (see Fig. S1 in the supplemental material) were chosen to be targeted by three single guide RNAs (sgRNAs; named PAM1, PAM2, and PAM3). The vulnerability of this gene was evaluated in both rich media (solid and liquid LB;. Fig. 2A to D) and defined media (solid 7H10 and liquid 7H9; Fig. 2E to H) in the presence or absence of anhydrotetracycline (ATc) 100 ng/mL, using the gene encoding the vulnerable MmpL3 protein (the essential *mmpL3* gene) as a positive control. This gene codes for the mycobacterial membrane MmpL3 protein, which is responsible for trehalose monomycolate transport through the cell inner membrane (23). In *M. tuberculosis* and *M. smegmatis*, it was shown that silencing *mmpL3* expression disrupts bacterial growth (20, 24), leading to accumulation of trehalose dimycolate (TDM) and cell death (25). We observed a similar growth pattern with or without ATc in liquid LB culture medium (Fig. 2B to D), although the levels of endogenous MsEPSPS were greatly reduced in *aroA*-knockdown cells with sgRNAs targeting sequences adjacent to any of the three motifs studied, as evaluated by immunoblot (Fig. 3A and B). By performing densitometric analysis, we found that the cellular levels of MsEPSPS in *aroA*-knockdown cells had a 5.7- (PAM1), 4.6- (PAM2), and 3.6- (PAM3) fold reduction, compared to the MsEPSPS levels of equivalent cultures grown without ATc. Interestingly, *aroA* knockdown directed to a sequence adjacent to the PAM sequence predicted to be of higher "strength" (20) (PAM1) resulted in the higher level of MsEPSPS reduction (5.7-fold), while the experiment targeting a sequence adjacent to the PAM sequence of predicted lower strength (PAM3) resulted in the lower level of MsEPSPS reduction observed (3.6-fold).

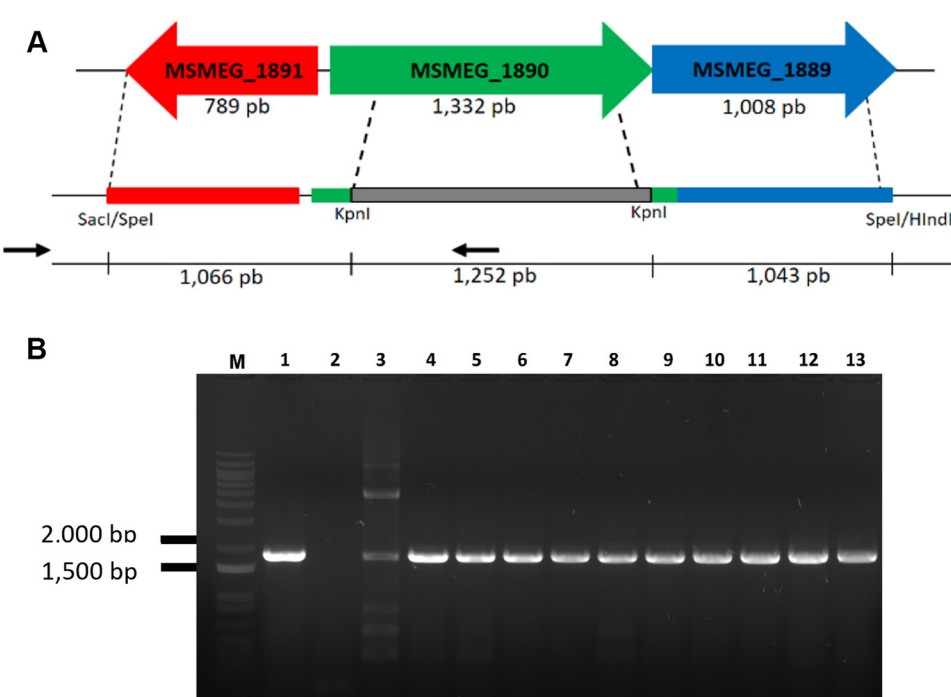

**FIG 1** *aroA*-deleted *M. smegmatis* cells are auxotrophic for aromatic amino acids. (A) Schematic representation of the allelic exchange event in the *aroA* locus. Two putative genes (MSMEG_1891 and MSMEG_1889) flank the *aroA* gene (MSMEG_1890) of *M. smegmatis*. The allelic exchange sequences (AESs) were designed to maintain possible transcriptional and translation regulatory sequences of these two genes. Most of the *aroA* gene sequence was replaced by a kanamycin resistance cassette (1,252 bp), which was also used as a selective marker for homologous recombination. The positions of primers used in PCRs described in panel B are indicated by black arrows. (B) Confirmation of *aroA* gene knockout by allelic exchange mutagenesis in merodiploid WT or control strain of *M. smegmatis* grown on defined 7H10 medium with AroAA supplementation. Double crossover (DCO) events on the original *aroA* locus in the *M. smegmatis* genome were confirmed by PCR amplification. Genomic DNA was extracted from selected white colonies and used as the templates for PCRs in the presence of a forward primer upstream the 5′ AES (allelic exchange sequence), outside the region of recombination, and an internal reverse primer (see Fig. 2A and Table 3). An amplicon of 1,813 bp is expected for allelic exchange mutants. Lane M: 1 kb plus DNA ladder (Invitrogen). Lanes 1 and 13: *M. smegmatis* cells grown on 7H10 medium without supplementation: *aroA*-deleted cells (A) from merodiploid strain containing an extra copy of WT *aroA* gene (positive controls). Lanes 2 to 12: *M. smegmatis* cells grown on defined 7H10 medium with AroAA supplementation. Lane 2: *M. smegmatis* mc² 155 genomic DNA (negative control). Lanes 3 to 4: *aroA*-deleted cells from merodiploid strain carrying an extra copy of WT *aroA* gene. Lanes 5 to 12: *aroA*-deleted cells from *M. smegmatis* strain containing an integrated copy of vector pNIP40/b (empty vector, pNIP40::Ø), without an extra copy of *aroA* gene.

In contrast to the similar growth curves observed in cultures grown on LB, we observed a marked decrease in bacterial growth at 24 h in the presence of ATc in cells grown in liquid (7H9) defined medium. This was observed for *aroA*-knockdown cells containing the CRISPRi system targeting sequences adjacent to any of the PAM motifs studied (PAM1, PAM2, and PAM3), indicating that *aroA* gene knockdown leads to a larger bacterial growth perturbation under these defined nutritional conditions (Fig. 2F to H). For the bacterial cells containing the CRISPRi targeting a sequence adjacent to PAM2 (Fig. 2C and G), there was no difference in growth for ATc+ and ATc− cultures after 24 h of growth in LB medium (Fig. 2C; $P > 0.999$), while the cultures differed at this same time point when grown in 7H9 medium (Fig. 2G; $P < 0.05$). On the other hand, for cells containing the CRISPRi system targeting sequences adjacent to PAM1 (Fig. 2B and F) and PAM3 (Fig. 2D and H), the growth difference between ATc-treated and untreated cultures after 24 h was significant in both nutritional conditions ($P < 0.01$). However, the difference in growth was much smaller in LB cultures compared to that in 7H9 cultures (mean difference in optical density [OD] of 0.17 [LB] and 0.97 [7H9] for PAM1 [Fig. 2B and F] and 0.29 [LB] and 0.78 [7H9] for PAM3 [Fig. 2D and

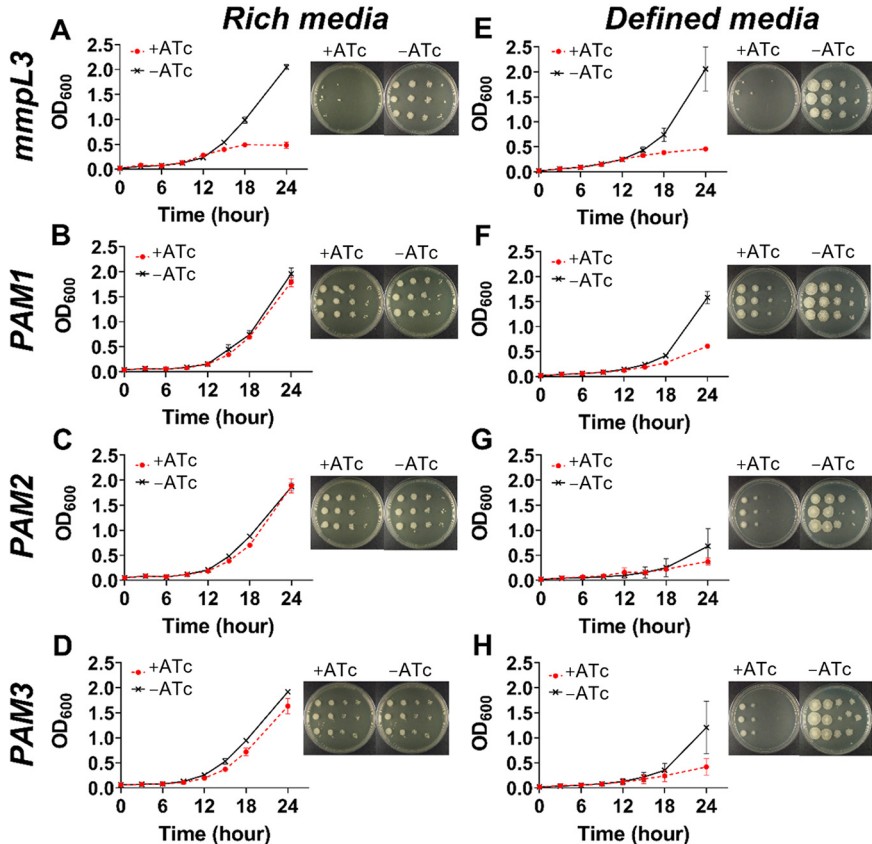

**FIG 2** Growth impairment of *AroA*-knockdown cells depends on growth conditions. (A to H) *M. smegmatis* growth curves and dilution spots in the presence or absence of anhydrotetracycline (ATc) (100 ng/mL). (A to D) Growth in rich media: liquid LB (curves) and solid LB (dilution spots shown in figure insets). (E to H) Growth in defined media: liquid 7H9 (curves) and solid 7H10 (dilution spots in figure insets). (A and E) Control gene *mmpL3*. (B to D and F to H) *aroA* gene knocked down using sgRNAs directed to locations adjacent to PAM1, PAM2, and PAM3 (B to D and F to H). Growth curves with and without ATc are different with $P < 0.01$ for all conditions except for PAM2 in rich (C; $P > 0.999$) and defined media (G; $P < 0.05$).

H]). We observed the same pattern for cultures grown on solid media, with differences between ATc-treated and untreated cultures grown in defined medium (7H10) larger than those between cultures grown in LB (Fig. 2B to H, insets). This dependency on medium composition for the growth perturbation found in *aroA*-knockdown cells was not observed in our control *mmpL3*-knockdown cells (Fig. 2A and E).

Next, we supplemented defined solid medium (7H10) with AroAAs (L-tryptophan plus L-phenylalanine plus L-tyrosine) and repeated the *aroA* knockdown using the CRISPRi system. These amino acids are synthesized inside cells by pathways that use the end product of the shikimate pathway, chorismate, as a starting compound (Fig. 4A). Interestingly, we rescued the growth pattern of the control sample (ATc−) when *aroA*-knockdown cells (ATc+) were grown in the presence of AroAAs (Fig. 4B to D).

**Identification of essential amino acid residues in *Ms*EPSPS.** To evaluate the essentiality of specific residues from *Ms*EPSPS, we constructed a set of merodiploid strains carrying an extra copy of a mutated *aroA* allele encoding D61W, E321N, or R134A *Ms*EPSPS mutants. To confirm the correspondence among residues from different enzymes, we aligned the EPSPS sequences from *E. coli*, *M. smegmatis*, and *M. tuberculosis* (Fig. 5A). The sequences from *M. smegmatis* and *M. tuberculosis* are 68% identical and 78% similar, while *M. smegmatis* and *E. coli* EPSPS proteins are 31% identical and 52% similar. We selected residues from *Ms*EPSPS located at the same positions of residues found to influence enzyme activity in *E. coli* and *M. tuberculosis* EPSPS orthologues. We found that the D61 residue from *Ms*EPSPS corresponds to the D49 and D54

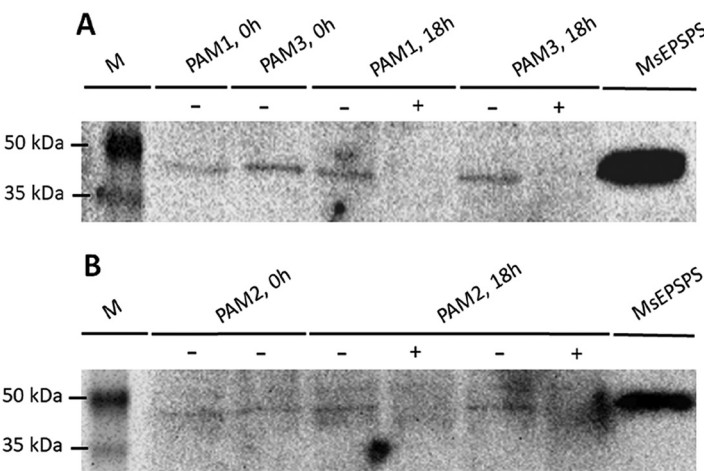

**FIG 3** *aroA*-knockdown cells are depleted of endogenous *Ms*EPSPS. Immunoblot with anti-*Mt*EPSPS polyclonal antibody of protein samples from *M. smegmatis* cells containing CRISPRi construct coding for sgRNA targeting sequence adjacent to PAM1, PAM2, or PAM3. *M. smegmatis* cells were grown on liquid LB medium. Lanes marked with (+) or (−) correspond to samples from cultures with or without induction with anhydrotetracycline (ATc), respectively. (A) Results for sgRNAs directed to PAM1- and PAM3-adjacent sequences. Lane M: ProSieve color protein markers (Lonza). Lanes corresponding to samples collected before induction with ATc (0 h) are from cells with CRISPRi system targeting PAM1- (PAM1, 0 h) or PAM3-adjacent (PAM3, 0 h) sequence. Lanes corresponding to samples collected 18 h after induction with (+) or without (−) ATc are also from cells with CRISPRi system targeting PAM1 (PAM1, 18 h) or PAM3 (PAM3, 18 h) adjacent sequence. MsEPSPS: purified *Ms*EPSPS as positive control. (B) Results for sgRNA directed to PAM2 adjacent sequence. Lane M: ProSieve color protein markers (Lonza). Samples were collected before induction (PAM2, 0 h) or 18h after induction (PAM2, 18 h) with (+) or without (−) ATc. MsEPSPS: purified *Ms*EPSPS as positive control.

residues from *Ec*EPSPS and *Mt*EPSPS sequences, respectively. A D49A substitution in *Ec*EPSPS led to a 24,000-fold reduction in the enzyme's specific activity (21), while a D54A or D54W substitution in *Mt*EPSPS was predicted to affect protein stability (22). We also found that two residues involved in the catalytic reaction of *Ec*EPSPS (21), R124 and D313, correspond to *Ms*EPSPS R134 and E321, respectively (Fig. 5A).

The mutant merodiploid strains were transformed with pPR27::KO_*aroA*, and three independent colonies of each strain were selected for DCO events. These experiments were performed in both LB and 7H10 agar media, conditions under which the WT *aroA* gene was found to be essential, as described above.

No viable colonies were rescued for R134A and E321N merodiploid mutants, indicating that both R134 and E321 are essential *Ms*EPSPS residues under the nutritional conditions evaluated (LB or 7H10 without AroAA supplementation). In contrast, the original WT *aroA* allele could be disrupted from the D61W merodiploid mutant strain. This was confirmed by PCR of genomic DNA extracted from white colonies selected on LB (Fig. 5B) or 7H10 (Fig. 5C) medium. We also sequenced the *aroA* allele after the allelic exchange process of all white colonies tested by PCR and shown in Fig. 5B (for both D61W and control WT merodiploid strains) as an additional confirmation that the *aroA* allele was effectively disrupted.

**Substitution of *Ms*EPSPS aspartate 61 by tryptophan (D61W) does not affect *M. smegmatis* growth.** Next, we evaluated whether the D61W mutation on *Ms*EPSPS affects bacilli growth. Growth curves were performed in LB medium for the *aroA*-deleted strain derived from the D61W merodiploid strain, together with the following controls: *aroA*-deleted strain derived from the WT merodiploid strain (extra copy of WT *aroA* gene), *M. smegmatis* mc$^2$ 155 wild-type strain, and pNIP40::Ø-integrated strain. We found that D61W mutation on *Ms*EPSPS does not affect bacilli growth (Fig. 5D).

**Production, purification, and identification of recombinant *Ms*EPSPS enzymes.** To evaluate how the D61W, R134A, and E321N substitutions on *Ms*EPSPS affect its kinetic properties, we produced the WT and mutant EPSPS proteins in recombinant form. The production of recombinant *Ms*EPSPS proteins (WT, D61W, R134A, and E321N) in the soluble

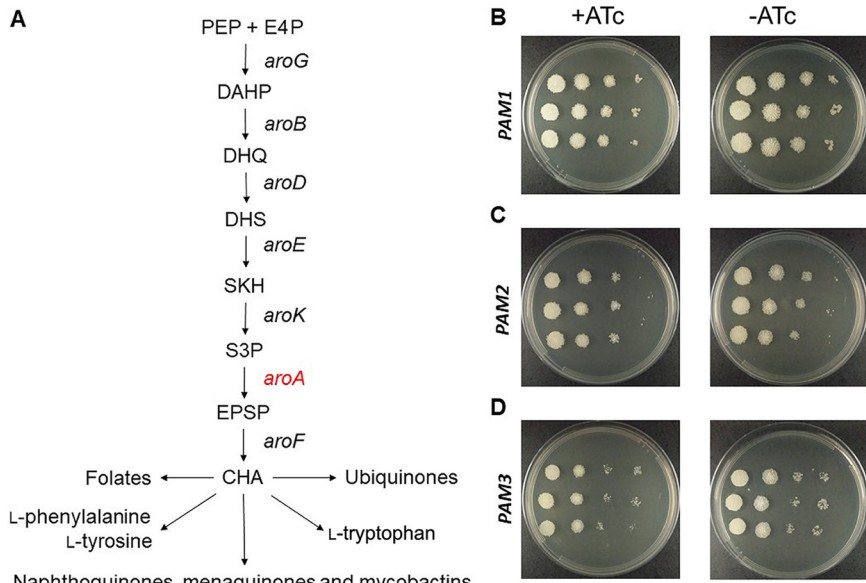

**FIG 4** Growth impairment of *aroA*-knockdown cells are rescued by AroAA supplementation. (A) Schematic representation of the shikimate pathway and the end products of pathways starting with chorismate. PEP: phosphoenolpyruvate; E4: ᴅ-erythrose 4-phosphate; *aroG*: gene encoding DAHP synthase (DAHPS); DAHP: 3-deoxy-d-arabino-heptulosonate-7-phosphate; *aroB*: gene encoding DHQ synthase (DHQS); DHQ: 3-dehydroquinate; *aroD*: gene encoding DHQ dehydratase (DHQD); DHS: 3-dehydroshikimate; *aroE*: gene encoding SKH dehydrogenase (SDH); SKH: shikimate; *aroK*: gene encoding SKH kinase (SK); S3P: shikimate-3-phosphate; *aroA*: gene encoding EPSP synthase (EPSPS); EPSP: 5-enolpyruvylshikimate-3-phosphate; *aroF*: gene encoding chorismate synthase (CS); CHO: chorismate. (B to D) Dilution spots of *aroA*-knockdown cells (+ATc) and control cells (−ATc) grown in defined solid medium (7H10) supplemented with AroAAs (ʟ-phenylalanine, ʟ-tyrosine and ʟ-tryptophan). +ATc: presence of anhydrotetracycline (100 ng/mL); −ATc: absence of anhydrotetracycline. (B) *aroA*-knockdown strain containing sgRNA directed to sequence adjacent to PAM1. (C) *aroA*-knockdown strain containing sgRNA directed to sequence adjacent to PAM2. (D) *aroA*-knockdown strain containing sgRNA directed to sequence adjacent to PAM3.

fraction was confirmed by SDS-PAGE, with an apparent molecular mass of 46 kDa. Homogeneous preparations were obtained using a 3-step protocol for both *Ms*EPSPS WT and D61W, whereas a 2-step protocol was employed for mutants R134A and E321N (Table 1; Fig. S3). Recombinant *Ms*EPSPS WT, D61W, E321N, and R134A mutants were submitted to trypsin digestion and peptides were analyzed by liquid chromatography-tandem mass spectrometry (LC-MS/MS). Coverage of approximately 90% was obtained for each protein with 85, 97, 80, and 74 unique peptides identified, respectively. Furthermore, it was possible to identify and validate all point mutations (Fig. S4 to S6).

**Kinetic parameters of WT and mutant EPSPS enzymes.** EPSPS enzymes are known to catalyze the transfer of the carboxyvinyl portion of PEP to the carbon-5 hydroxyl group of S3P, forming the EPSP product and releasing inorganic phosphate ($P_i$). To determine the kinetic parameters of WT and mutant enzymes, we performed a coupled assay using *Mt*PNP and MESG. The dependence of initial velocity on PEP as a variable substrate at fixed-saturating S3P concentration (see Table S1) followed hyperbolic Michaelis-Menten kinetics. The apparent steady-state kinetic parameters for WT and mutant *Ms*EPSPS enzymes are presented in Table 2.

## DISCUSSION

Target validation is a required part of any effort to develop new chemotherapeutic agents based on rational drug design. Essentiality for bacterial growth and/or survival is a critical feature of a target, as the chemical inhibition of nonessential gene products are not expected to kill the infective agent and hence to achieve the desired therapeutic outcome. Here, using *M. smegmatis* as a mycobacterial model organism, we show that the *aroA* gene, which codes for the enzyme 5-enolpyruvylshikimate-3-phosphate

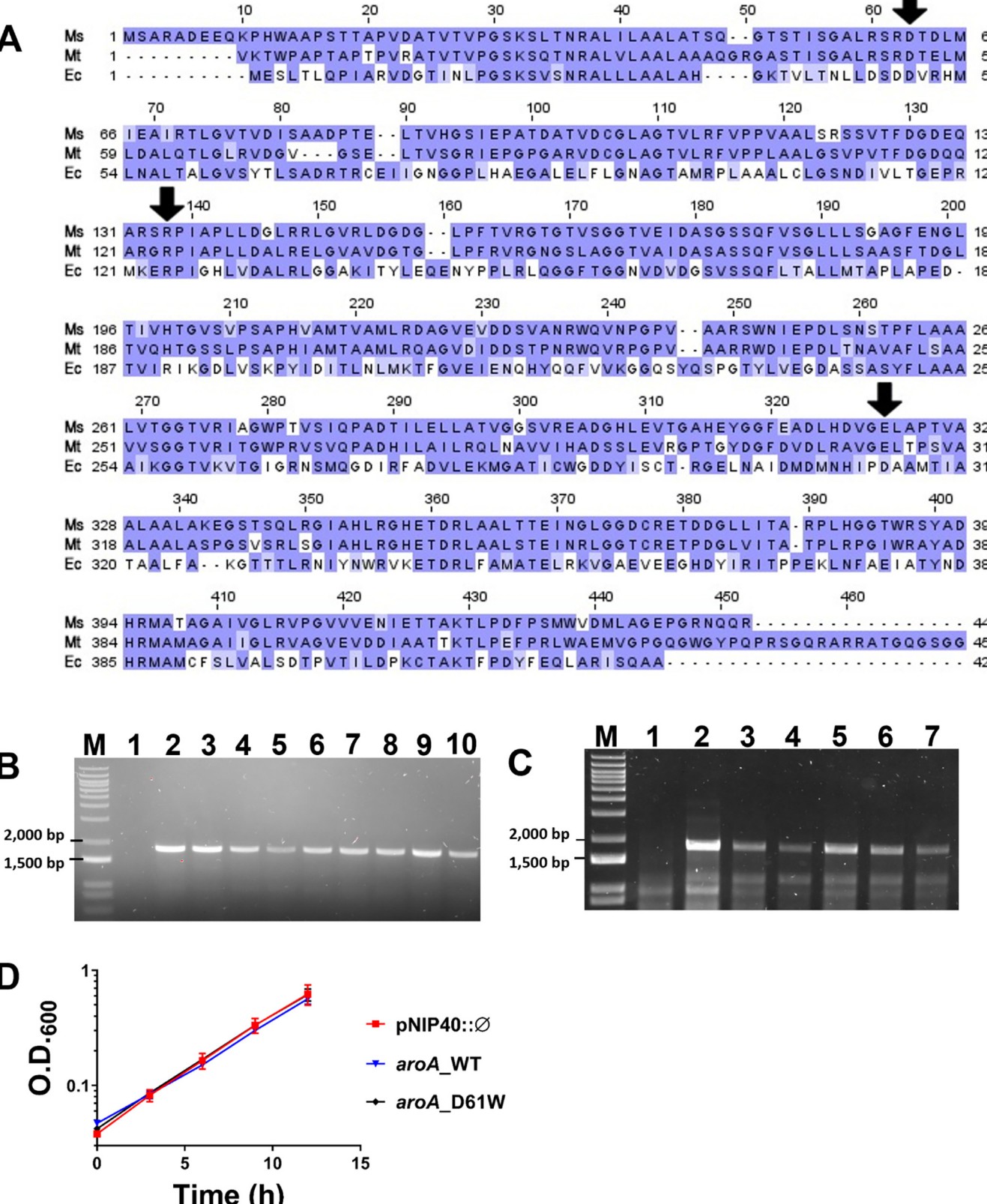

**FIG 5** The D61W substitution in *Ms*EPSPS does not impair mycobacterial survival and growth *in vitro*. (A) Sequence alignment of EPSPS enzymes from *M. smegmatis* mc² 155 (Ms), *M. tuberculosis* H37Rv (Mt), and *E. coli* CVM N33429PS (Ec). A multiple alignment for these proteins was made using Clustal Omega and then visualized and colored in Jalview. The color code represents the level of conservation of each amino acid, where darker shades of purple represent a higher conservation level. The enzymes from *E. coli* and *M. tuberculosis* have 52% and 78% positives and 31% and 68% of identity when

**TABLE 1** Purification yield of recombinant *Ms*EPSPS enzymes

| *Ms*EPSPS enzyme[a] | Column[b] | Protein concn (mg/mL) | Eluted vol (mL) | Total protein (mg) | Yield (%) | Homogeneity (%) |
|---|---|---|---|---|---|---|
| WT | First | 13.4 | 25 | 334.4 | 20.7 | 97.3 |
| | Last | 3.6 | 19 | 69.4 | | |
| D61W | First | 6.4 | 25 | 160.7 | 8.5 | 96.8 |
| | Last | 0.7 | 20 | 13.7 | | |
| R134A | First | 20.4 | 25 | 510.7 | 11.2 | 98.4 |
| | Last | 3.0 | 19 | 57.1 | | |
| E321N | First | 13.2 | 25 | 330.0 | 17.3 | 100 |
| | Last | 0.6 | 96 | 57.2 | | |

[a]Recombinant wild-type (WT) or mutant EPSPS enzymes from *M. smegmatis*.
[b]Chromatographic column used in the first or last step of the purification protocol.

synthase (EPSPS) from the shikimate pathway, is essential only when sufficient amounts of L-tryptophan, L-phenylalanine, and L-tyrosine (AroAAs) are not available (Fig. 1). We are not aware of previous attempts to generate *aroA*-deficient strains in *M. smegmatis*. Considering the pivotal role of the shikimate pathway of supplying cells with the end product chorismate, this can be considered a surprising result. Apart from the pathways leading to L-tryptophan, L-phenylalanine, and L-tyrosine, chorismate is the starting compound for the *p*-aminobenzoate branch of folates, for the synthesis of ubiquinones from *p*-hydroxybenzoate, and for the synthesis of isochorismate, leading to naphthoquinones, menaquinones, and mycobactins (Fig. 4A).

The entire shikimate pathway was deemed essential in the related *M. tuberculosis* (14). This conclusion was based on the inability to retrieve viable *M. tuberculosis* cells deficient in *aroK*, the gene that codes for the preceding enzyme in this pathway, shikimate kinase. Interestingly, viable cells could not be retrieved even with the addition of a supplement containing AroAAs, *p*-hydroxybenzoate, *p*-amino-benzoic acid, and 2,3-dihydroxybenzoate (14). These metabolites can be synthesized in mycobacteria from the end product of the shikimate pathway, chorismate, and are part of known metabolic pathways that use chorismate as a starting compound. Noteworthy, in the absence of redundant activities, *aroK* and *aroA* deletion are expected to have the same impact in terms of chorismate production.

Nevertheless, our findings are not an exception if we compare them with those of similar studies conducted with other bacteria. For example, *aroA*-deleted strains from *Salmonella enterica* serovar *Infantis* (17), *Pseudomonas aeruginosa* (18), and *Burkholderia glumae* (19) were found to be auxotrophic for the three AroAAs (L-tryptophan, L-phenylalanine, and L-tyrosine). For other bacterial species, the supplementation with aromatic compounds was also sufficient to rescue viable *aroA*-deleted cells, but the "aromix" included additional components other than AroAAs. For instance, in *Aeromonas hydrophila* (26), *Aeromonas salmonicida* (27), *Shigella flexneri* (28), and *Salmonella enterica* serovar Typhimurium (29), viable

**FIG 5** Legend (Continued)

aligned to the EPSPS from *M. smegmatis*, respectively. Amino acids indicated by black arrows were the ones chosen for mutagenesis. (B and C) PCR confirmation of double crossover (DCO) events leading to deletion of the original *aroA* allele in the *M. smegmatis* genome from merodiploid strains carrying an extra copy of the WT or mutated *aroA* gene encoding D61W EPSPS mutant. Genomic DNA was extracted from selected white colonies and used as the templates for PCRs in the presence of a forward primer upstream the 5' AES (allelic exchange sequence), outside the region of recombination, and an internal reverse primer (see Table 3). The PCRs were performed on white colonies selected on LB medium (B) or 7H10 medium without supplementation with AroAAs (C). An amplicon of 1,813 bp was expected for allelic exchange mutants. (B) Lane M: 1 kb plus DNA ladder (Invitrogen). Lane 1: *M. smegmatis* mc² 155 genomic DNA (negative control). Lanes 2 to 8: *aroA*-knockout cells obtained from a merodiploid strain carrying an extra copy of WT *aroA* gene. Lanes 9 to 10: *aroA*-knockout cells obtained from a merodiploid strain carrying an extra copy of *aroA* mutated that encodes the D61W *Ms*EPSPS protein. (C) Lane 1: *M. smegmatis* mc² 155 genomic DNA (negative control). Lanes 2 to 7: *aroA*-deleted cells from merodiploid strains carrying an extra copy of WT *aroA* gene (2–6) or a mutant *aroA* gene encoding D61W EPSPS protein (7). (D) Control strains and a strain carrying an altered *aroA* sequence that encodes D61W EPSPS mutant (*aroA*_D61W) and that have had the original *aroA* allele deleted were grown for 12 h in LB medium, under aerobic conditions. Aliquots were taken every 3 h for optical density measurement at 600 nm (OD$_{600}$). pNIP40::Ø: strain carrying the original *aroA* allele and an empty copy of the pNIP40/b vector integrated into the mycobacteriophage Ms6 chromosomal integration site; *aroA*_WT: strain having the original *aroA* allele deleted but carrying another copy of the WT *aroA* gene integrated. Error bars are standard deviation (SD) of three biological replicates. Growth curves from control strains pNIP40::Ø and *aroA*_WT are not statistically different compared to *aroA*_D61W growth curve at any time point ($P > 0.99$ for all comparisons at 0 to 9 h; $P = 0.49$ for *aroA*_WT compared to *aroA*_D61W at 12 h; $P > 0.99$ for pNIP40::Ø compared to *aroA*_D61W at 12 h).

**TABLE 2** Apparent steady-state kinetic parameters for *Ms*EPSPS enzymes

| *Ms*EPSPS enzyme[a] | $K_m$ ($\mu$M) | $k_{cat}$ (s$^{-1}$) | $k_{cat}/K_m$ (M$^{-1}$s$^{-1}$) |
|---|---|---|---|
| WT | 88 $\pm$ 11 | 0.5530 $\pm$ 0.0185 | 6.28 E+03 $\pm$ 813 |
| D61W | 1,014 $\pm$ 975 | 0.1075 $\pm$ 0.0608 | 1.06 E02 $\pm$ 118 |
| R134A | 3,676 $\pm$ 1,007 | 0.1843 $\pm$ 0.0330 | 5.01 E01 $\pm$ 16 |
| E321N | 3,081 $\pm$ 808 | 0.4378 $\pm$ 0.0469 | 1.42 E02 $\pm$ 40 |

[a]Recombinant EPSPS enzymes from *M. smegmatis*: WT and point mutants. S3P was used at saturating concentrations (see Table S1 in the supplemental material) and PEP as a variable substrate in the enzymatic assay. All reactions were performed in duplicate.

*aroA*-deleted cells were rescued after supplementing growth cultures with the three AroAAs and *p*-aminobenzoic acid (pAB). Some variations on the aromix composition were employed to rescue viable *aroA*-deleted cells from other bacterial species. In *Pasteurella multocida*, it contained AroAAs, 2,5-dihydroxybenzoic acid (DHB), and *p*-hydroxybenzoic acid (30), and in *Bordetella pertussis*, the aromix contained AroAAs, DHB, and pAB (31). Although the shikimate pathway is generally considered essential for bacteria, plants, and fungi, clearly there are still unsolved issues and discrepancies when we consider its essentiality in terms of metabolic requirements from downstream pathways.

Next, we addressed the issue of target vulnerability. A target should be not only essential but also vulnerable; otherwise, chances are low to develop bioactive compounds that effectively kill or impart growth defects on infective agents. To evaluate *Ms*EPSPS vulnerability, we performed gene knockdown experiments using a CRISPRi system developed for mycobacteria (20). Using an in-house Python script, we selected target sequences adjacent to PAM motifs whose repression strengths were characterized previously (20). The experiments were conducted in both rich (LB; Fig. 2A to D) and defined (7H9 or 7H10; Fig. 2E to H) media, with markedly different results. As expected, the sgRNA control targeting the *mmpL3* gene was found to be vulnerable in both nutritional conditions, in either liquid or solid medium. Silencing *mmpL3* caused a cessation of bacterial growth by 15 h, and using the drop method on plates, we have observed a reduction of at least 1,000-fold in the CFU counting (Fig. 2A and E). Differently from MmpL3, *Ms*EPSPS was found to be vulnerable only in defined medium, irrespective of PAM's repression strength. Silencing *aroA* gene caused an impairment of the bacterial growth at 24 h but not a complete cessation (Fig. 2F to H). This suggests that the abrupt reduction on endogenous *Ms*EPSPS levels observed for *aroA*-knockdown cells (Fig. 3) does not cause bacterial death but, rather, growth impairment. Moreover, supplementation with AroAAs is sufficient to rescue the growth impairment of *aroA*-knockdown strains (Fig. 4B to D).

In contrast to what we found in defined medium (liquid 7H9 or solid 7H10), we observed growth patterns for *aroA*-knockdown cells grown on rich medium (liquid or solid LB) similar to those for control cells (Fig. 2A to D). We found only a subtle change (although significant) in growth curves at 24 h in rich medium for *aroA*-knockdown cells containing the CRISPRi system targeting sequences adjacent to PAM1 and PAM3 motifs (Fig. 2B and D). However, in both growth conditions (LB or 7H9/7H10), no viable *aroA*-deleted cells could be retrieved without AroAA supplementation. Our results support the notion that target essentiality and vulnerability are strongly dependent on nutritional context and care must be taken to extrapolate the information gathered from *in vitro* models to the context of the disease.

Finally, we evaluated the essentiality of selected amino acid residues from *aroA*-coded *Ms*EPSPS under growth conditions devoid of AroAA supplementation. The selection of these residues was based on a previous experimental work on EPSPS from *E. coli* (*Ec*EPSPS) (21) and on computational studies of *Mt*EPSPS from *M. tuberculosis* (22). It was shown that substitution of the aspartic acid 49 (D49) to alanine in *Ec*EPSPS (position D61 in *Ms*EPSPS, see Fig. 5A) leads to a reduction of 24,000 times in the specific activity of the enzyme. The reasons for that are still unclear, but the authors hypothesized an indirect effect on the lysine 22 (K22) residue, which is known to participate directly in catalysis (21). In addition, *in silico* predictions using *Mt*EPSPS suggested that

substituting the aspartic acid 54 (D54) residue (the same D61 position in *Ms*EPSPS) to alanine (D54A) or tryptophan (D54W) should cause a significant impact on the protein stability and, consequently, a negative impact on the enzyme's activity (22). On the other hand, arginine 124 (R124) and aspartic acid 313 (D313) (R134 and E321 in *Ms*EPSPS, respectively; Fig. 5A), which are near the PEP binding site, are directly involved in the catalytic reaction. The catalytic activities of R124A and D313E mutants were found to be 5,000 and 20,000 times lower, respectively, than the WT enzyme activity, implying a critical role for them in the normal functioning of *Ec*EPSPS (21).

The strategy adopted to evaluate the essentiality of selected residues of *Ms*EPSPS was 2-fold: first, we evaluated experimentally the predicted impact of selected mutations (D61W, R134A, and E321N) on the activity of recombinant *Ms*EPSPS, and then we constructed three merodiploid strains containing extra copies of *aroA* encoding point mutants of *Ms*EPSPS (D61W, R134A, and E321N) and tried to remove the original WT *aroA* copy in gene disruption experiments.

For the first part of our approach, we produced and purified the recombinant *Ms*EPSPS WT enzyme and mutants D61W, R134A, and E321N. The kinetic properties of the WT and the three mutants were measured and compared (Table 2). The $K_m$ for the substrate PEP in mutant forms of *Ms*EPSPS increased from 11.5 to 42 times compared to that of the WT enzyme. These results suggest an increased overall dissociation constant for PEP binding to mutant proteins at fixed-saturating concentrations of S3P. The impact on enzyme turnover ($k_{cat}$) ranged from 1.2-fold (E321N) to 5.1-fold (D61W) decrease (Table 2). Accordingly, more pronounced effects on the catalytic efficiencies ($k_{cat}/K_m$) of mutants were observed. We found a reduction of 44- (E321N), 59- (D61W), and 125-fold (R134A) in $k_{cat}/K_m$ for these enzymes. These reductions in the apparent second-order rate constants suggest lower association rate constants for PEP substrate binding to *Ms*EPSPS. Therefore, we can conclude that mutations in these specific residues affect the catalytic efficiency of *Ms*EPSPS, although to a lesser extent than expected, based on previous studies with orthologs (21, 22).

After confirming and quantifying the impact of D61W, R134A, and E321N substitutions on *Ms*EPSPS enzymatic efficiency, we evaluated the essentiality of those amino acid residues by constructing three merodiploid strains containing extra copies of *aroA* encoding the corresponding point mutants of *Ms*EPSPS. As described, we were unable to obtain viable cells deficient in the original WT *aroA* gene from merodiploid strains containing as the extra copy *aroA* sequences coding for R134 and E321 *Ms*EPSPS mutants. This indicates that R134 and E321 amino acids are indeed essential, which is consistent with the reduction observed in enzyme efficiency for these mutants and their catalytic role on EPSPS orthologs. In contrast, the results obtained with D61W merodiploid strain indicate that D61 is nonessential. Viable cells deficient in the original WT *aroA* gene were obtained from this merodiploid strain in both LB and 7H10 media (Fig. 5B and C). We further characterized this *aroA*-deleted strain and evaluated whether *Ms*EPSPS D61W substitution would result in any impact on bacterial growth. Again, we did not detect any growth defect (Fig. 5D). These results suggest that despite the 59-fold reduction in enzyme efficiency ($k_{cat}/K_m$) determined *in vitro*, the impact of the mutation on the noncatalytic D61 residue (D61W) inside cells may be milder. It is possible that the dynamic features of the enzyme supposedly affected by this substitution (22) might also depend strongly on the medium composition. It is known that differences in cytosol and medium compositions of *in vitro* reactions can greatly affect the activities of different enzymes (32).

Our study raises a warning about using EPSPS as a target for drug development. As *aroA*-deficient cells were found to be auxotrophic for AroAAs and normal growth curves could be rescued for *aroA*-knockdown cells by AroAA supplementation, we predict that antimicrobial agents acting as EPSPS inhibitors would become ineffective if bacilli have access to external supplies of AroAAs. However, it is still unknown whether, in the context of the disease, the host microenvironment of infecting Mtb bacilli supplies their requirements for AroAAs. Data obtained from intracellular infecting Mtb cells indicate that at least phenylalanine and tyrosine need to be synthesized *de novo* by bacterial cells (33). On the other hand, in another study (34), an auxotrophic mutant strain of *M. tuberculosis* for

**TABLE 3** List of primers used in this study

| Primer | Primer sequence | |
|---|---|---|
| | Primer F | Primer R |
| Primers used in the mutation experiments | | |
| aroA_D61W | GATCATGAGGTCGGTCCAGCGGCTGCGCAGCGC | GCGCTGCGCAGCCGCTGGACCGACCTCATGATC |
| aroA_R134A | AGCGATGGGCGCTGACCTGGCCTGTTCGTC | GACGAACAGGCCAGGTCAGCGCCCATCGCT |
| aroA_E321N | CGGTCGGCGCGAGATTACCCACGTCGTGC | GCACGACGTGGGTAATCTCGCGCCGACCG |
| Primers used for allelic exchange mutagenesis | | |
| aroA_WT | TTTCATATGAGTGCACGCGCGGACGA | TTTAAGCTTTCTAGATTCAACGCTGTTGATTCCTCCCC |
| AES_Up | TTTGAGCTCACTAGTATCGCATCGATGACCGCG | TTTGGTACCCCGCTGATCGTGGAGGTG |
| AES_De | TTTGGTACCGGGGTCGTCGTCGAGAACAT | TTTAAGCTTACTAGTGAGCGCGCACTCCGGATC |
| Primers used for double crossover amplification | | |
| DCO | GAAGAAGTCGTGAGTGCCGT | GTTTTCCCGGGGATCGCAGT |
| Primers used for phsp60 + aroA sequencing | | |
| PCR_Phsp60/CO_aroA | CTTTGATCGGGGACGTCTG | CGGGGAGGAATCAACAGCGTTGA |
| SEQ_CO_aroA/Phsp60 | GGGGTGCGCCTCGACGGC | GAGGTCGACGATTCGGTG |
| Primer used for CRISPRi sequencing | | |
| SEQ_CRISPRi_1834 | | TTCCTGTGAAGAGCCATTGATAATG |
| Oligonucleotides used for sgRNA[a] | | |
| PAM1_NNAGCAT | GGGA**G**ACCTCGACGCCCGCGTCGC | AAACGCGACGCGGGCGTCGAGGTC |
| PAM2_NNAGGAT | GGGA**G**CCCTGCGAGGTGGCCAGCGCCG | AAACCGGCGCTGGCCACCTCGCAGGGC |
| PAM3_NNAGCAG | GGGA**A**CCCCGAGCCGGCGCAGGCCGT | AAACACGGCCTGCGCCGGCTCGGGGT |
| mmpL3 | GGGA**G**CGACAGATGGCTGCCCTCGTC | AAACGACGAGGGCAGCCAGTCTGTCGC |

[a]Transcription start sites of coded sgRNAs are in bold.

tryptophan (defined mutation on *trpD* gene from the tryptophan biosynthesis pathway) was severely attenuated, with more than 90% of bacterial cells killed after 10 days of infection of bone marrow-derived macrophages. This same strain was completely avirulent after mice infection, indicating that at least tryptophan is not available in sufficient amounts for Mtb bacilli under these conditions. Evaluation of *aroA* gene essentiality and vulnerability in experimental models that resembles more closely the physiological environment experienced by infecting bacilli in TB would be helpful to clarify this point. Moreover, it will be important to evaluate whether *aroA*-deficient and *aroA*-knockdown bacilli from *M. tuberculosis* behave as we have found here for *M. smegmatis*.

Under defined growth conditions devoid of AroAA supplementation, we found that *aroA* is both essential and vulnerable. Moreover, taken together, our mutational studies indicate that *aroA* essentiality in these growth conditions is causally linked to EPSPS activity. As pointed out by some of us, establishing a causal link between gene essentiality and the biological function of its protein product under scrutiny should be an indispensable step in target validation for drug development (35). This view is reinforced by the growing number of proteins found to exhibit multiple and unrelated tasks, the so-called moonlighting proteins (36).

## MATERIALS AND METHODS

**Bacterial strains, growth conditions, and transformation.** *E. coli* DH10B strain was used for all cloning procedures and routinely grown in LB medium (broth and agar) at 37°C. *M. smegmatis* mc² 155 strain (37) was kindly provided by William R. Jacobs, Jr., Albert Einstein College of Medicine, NY, USA. *M. smegmatis* was used for gene disruption and knockdown experiments, grown in LB (tryptone, yeast extract, NaCl, and agar) medium, Difco Middlebrook 7H9 broth (Becton, Dickinson - BD), supplemented with 0.05% (vol/vol) Tween 80 (Sigma-Aldrich), and 0.2% (vol/vol) glycerol (Merck), or Difco Middlebrook 7H10 agar (BD), supplemented with 0.5% (vol/vol) glycerol. Wherever required, the following antibiotics or small molecules were used: 50 $\mu$g/mL ampicillin (Amp; Sigma-Aldrich), 25 $\mu$g/mL kanamycin (Kan; Sigma-Aldrich) for culturing recombinant *E. coli* strains. Also, we used 25 $\mu$g/mL Kan, 50 $\mu$g/mL hygromycin (Hyg; Invitrogen), 100 ng/mL anhydrotetracycline (ATc; Sigma-Aldrich), and 50 $\mu$g/mL of each amino acid L-tryptophan (FisherBiotech), L-phenylalanine (Sigma-Aldrich), and L-tyrosine (Sigma-Aldrich) for culturing *M. smegmatis* strains. *E. coli* strains were routinely transformed by electroporation using cuvettes of 0.2 cm, with a 200 $\Omega$ resistance, 25 $\mu$F capacitance, and pulses of 2.25 kV for 3 s. For *M. smegmatis* strains, the resistance was changed to 1,000 $\Omega$ and the pulses to 2.5 kV (38). All primers used in this study are listed in Table 3.

Microbiology
Spectrum

**Construction of plasmids for recombinant protein production.** The WT *aroA* gene (MSMEG_1890), predicted to encode an EPSPS, was amplified by PCR using aroA_WT_Primer F and aroA_WT_Primer R (Table 3), 25 ng of genomic DNA of *M. smegmatis*, and 10% dimethyl sulfoxide (DMSO). Genomic DNA was extracted and purified according to a published protocol (39). The PCR product of 1,354 bp was gel purified, cloned into the pCR-Blunt (ThermoFisher) vector, and subcloned into the pET-23a(+) (Novagen) expression vector, using NdeI and HindIII restriction sites. The pET-23a(+)::*aroA*(WT) recombinant vector was also used as the template for mutagenesis reactions. Three different mutations (D61W, R134A, and E321N) were incorporated into the gene sequence using the QuikChange XL site-directed mutagenesis kit (Stratagene), along with a set of several primers: primer pairs F and R of *aroA*_D61W, *aroA*_R134A, and *aroA*_E321N (Table 3). Recombinant clones were confirmed by DNA sequencing.

**Construction of the plasmid for allelic exchange mutagenesis.** The genomic flanking sequences of *aroA* gene from *M. smegmatis* were PCR amplified to serve as allelic exchange substrates (AESs) for gene disruption. The upstream flanking sequence (1,066 bp) was amplified using oligonucleotides AES_Up_Primer F and AES_Up_Primer R, which contain SacI/SpeI and KpnI restriction sites, respectively (Table 3). The amplicon obtained (AES_Up) included 164 bp of the 5′-end of *aroA* gene. The downstream flanking sequence (1,043 bp) was amplified using oligonucleotides AES_Dw_Primer F and AES_Dw_Primer R, which contain KpnI and SpeI/HindIII restriction sites, respectively (Table 3). The resulting amplicon (AES_Dw) included 109 bp of the 3′-end of *aroA* gene. The AES_Up sequence was cloned into the pUC19 vector using restriction sites for SacI and KpnI, followed by the AES_Dw insertion using KpnI and HindIII restriction sites. Both AES sequences were confirmed by DNA sequencing. The vector was then digested with KpnI, the cohesive ends were filled with *Pfu* DNA polymerase (QuatroG P&D), and the resulting plasmid was dephosphorylated with calf intestinal phosphatase (CIP; Invitrogen). A 1.2 kb kanamycin resistance cassette from the pUC4K vector was inserted between the AESs. Finally, the whole construction was cut out from pUC19 with SpeI and inserted into the SpeI site of pPR27*xylE* vector (40), yielding the plasmid pPR27::KO_*aroA* (Table 3), used to perform allelic replacement.

**Construction of integrative plasmids for generation of merodiploid strains.** The WT and three mutant *aroA* sequences (D61W, R134A, and E321N) were transferred from the pET-23a(+) vector to the pMVHG1 shuttle vector (41) using the NdeI and HindIII restriction sites. Each gene sequence (WT and mutants) was ligated downstream to the heat shock promoter P$_{hsp60}$. Then, the P$_{hsp60}$::*aroA* sequences were cut out with XbaI, gel purified, and inserted into the XbaI-dephosphorylated site of the pNIP40/b plasmid, yielding four different integrative plasmids used to generate merodiploid strains.

**Construction of gene KD plasmids.** The vulnerability of the *aroA* gene was evaluated using the CRISPRi system, developed by Rock and colleagues (20). Three single guide RNAs (sgRNAs) were designed to target different regions of the *aroA*-coding sequence (see Fig. S1 in the supplemental material). They were designed to bind the nontemplate (NT) strand of the *aroA* gene in regions where three different PAM (protospacer-adjacent motif) sequences (Table 3) were identified using an in-house script written in Python and made publicly available in the GitHub repository (https://github.com/Eduardo-vsouza/sgRNA_predictor). An sgRNA targeting the *mmpL3* (MSMEG_0250) gene was used as a positive control of knockdown (KD) experiments.

The PLJR962 vector backbone was linearized by BsmBI digestion and gel purified. Two partially cDNA oligonucleotides (20 to 25 nucleotides [nt] in length) were designed to code for the target binding portion of each sgRNA (PAM1, PAM2, and PAM3 oligonucleotides F and R; Table 3). To ensure high transcription efficiency, all oligonucleotides were designed in such a way that sgRNAs transcribed from them initiated with an "A" or "G" at their 5′ ends (Table 3, nucleotides in bold). After annealing (95°C for 5 min; decrease 0.1°C/s until reaching 25°C), oligonucleotides retain single-stranded 5′ protruding ends that are complementary to the cohesive ends of BsmBI-digested PLJR962 vector. The ligation of annealed oligonucleotides into the vector backbone using T4 DNA ligase (23°C for 16 h) was confirmed by BsmBI digestion and DNA sequencing.

**Construction of *aroA* merodiploid strains and *aroA* gene disruption under three different nutritional conditions.** Gene disruption experiments were performed in strains carrying either an extra copy of *aroA* (WT or mutant merodiploid strains) or the empty vector pNIP40/b (pNIP40::Ø-integrated strain), as reported previously (35). Merodiploid strains carry an extra copy of WT or mutant *aroA* gene (encoding R134A, E321N, or D61W *Ms*EPSPS mutants) integrated into a specific chromosomal region (overlapping the 3′ end of the tRNA$^{Ala}$ gene) containing a mycobacteriophage Ms6 chromosomal integration site (42).

To prepare the pNIP40::Ø-integrated and merodiploid strains, 100 to 300 ng of the empty vector (pNIP40::Ø) or of each pNIP40/b construct was used to transform electrocompetent *M. smegmatis* cells (200 μL) by electroporation. After 3 days of incubation with hygromycin at 37°C, one colony from each transformation was grown in 5 mL of LB medium, and electrocompetent cells were prepared again.

Next, we transformed all strains with a plasmid carrying the allelic exchange substrate (pPR27::KO_*aroA*) and selected three to five independent transformant colonies for each strain in each of the nutritional conditions evaluated (LB and 7H10 solid media for mutant merodiploid strains or LB, 7H10, and 7H10 + AroAAs solid media for WT and pNIP40::Ø-integrated strains). All the steps required to select for DCO events were performed for each transformant independently. Transformants were selected using hygromycin (pNIP40/b resistance marker) and kanamycin (pPR27*xylE* resistance marker) at 32°C (permissive temperature). Colonies were also tested for the presence of the *xylE* reporter gene (*xylE*$^+$) with the addition of a drop of 1% catechol solution (Sigma-Aldrich). Three to five catechol-positive yellow colonies (Kan$^R$, Hyg$^R$, XylE$^+$) were grown separately in liquid medium (LB, 7H9, or 7H9 plus AroAAs), with kanamycin and hygromycin, at 32°C and 180 rpm, until reaching an optical density (OD) of 0.6 to 1.0 at 600 nm. Approximately $1 \times 10^7$ CFU were then plated on solid medium (LB, 7H10, or 7H10 plus AroAAs) plus kanamycin plus hygromycin plus 2% sucrose counterselection plates, in triplicate, and incubated at 39°C for 5 days. The inoculum was determined by plating each culture on the same medium, but in the absence of sucrose, at the permissive temperature

(32°C), for 7 days. Plates were analyzed for the presence of recombinant catechol-negative white colonies (Kan$^R$, Hyg$^R$, XylE, Suc$^R$) with a drop of catechol. DCO events resulting in removal of the original aroA allele were confirmed by PCR using genomic DNA extracted from each white colony selected. Amplification reactions were performed with DCO_Primer F and DCO_Primer R (Table 3). DCO_Primer F anneals upstream from aroA gene, outside the recombination region, while DCO_Primer R anneals inside the kanamycin resistance cassette (Fig. 1A). An amplicon of 1,813 bp in length was obtained from allelic exchange mutants that underwent a DCO event (Fig. 1B). The genomic DNA of WT M. smegmatis was used as a negative control. The aroA sequences from WT and mutant colonies were confirmed by DNA sequencing. The consensus sequences were assembled using Staden Package 4 (version 1.6). Sequences of PCR and sequencing primers are shown in Table 3.

**Aerobic growth curves of aroA gene-deleted strains.** The M. smegmatis strains that were viable after the deletion of the WT aroA chromosomal copy were grown in LB medium plus kanamycin plus hygromycin until reaching the early log phase (OD$_{600}$ ≈ 0.2). Cultures were then diluted into fresh LB medium with antibiotics to a theoretical OD$_{600}$ of 0.02 and divided (16 mL) into three conical tubes of 50 mL. Cultures were further incubated for 12 h, at 37°C, under shaking (180 rpm) and aerobic conditions. Aliquots of 1 mL were taken every 3 h and OD$_{600}$ was monitored. Results were expressed as mean ± standard deviation (SD) of three biological replicates.

**Gene knockdown by CRISPRi.** Knockdown experiments using CRISPRi were performed on liquid and solid media. First, electrocompetent M. smegmatis cells were transformed with PLJR962 constructs containing the sgRNA coding sequences (see section "Construction of gene KD plasmids"), and transformants were selected on solid LB with kanamycin. After 3 days of incubation, three isolated colonies were cultivated in 5 mL of LB for 48 h, at 37°C, under shaking (180 rpm). Cultures were then diluted (1:200) in LB (100 mL) containing kanamycin and further incubated to reach an OD$_{600}$ of 0.2 (for growth curve) or 0.6 (for drop method on plates). For gene KD in liquid medium, cultures were further diluted in fresh medium (OD$_{600}$ ≈ 0.02) containing kanamycin, equally divided (16 mL) in three conical tubes of 50 mL, with or without ATc, and grown for 24 h at 37°C. Samples (1 mL) were taken every 3 h. Results were expressed as mean ± SD of three biological replicates. Two-way analysis of variance (ANOVA) followed by Bonferroni's multiple-comparison test was performed using GraphPad Prism version 7.0.0 for Windows, GraphPad Software, San Diego, California, USA. For gene KD in solid medium, drops of 5 $\mu$L were plated on solid LB containing kanamycin, with or without ATc. The first spot contained approximately 5,000 cells, and the other three subsequent spots were 10-fold serially diluted. Plates were incubated for 3 to 4 days at 37°C. A negative (PLJR962::Ø) and a positive (PLJR962::mmpL3) control were employed for each condition. Additionally, KD experiments were performed in absence of AroAAs presence or absence of AroAAs.

**Protein extraction of M. smegmatis.** For each sample of gene KD in liquid medium, total protein was extracted in 0 h and 18 h, as described previously (43, 44). Cellular pellets were washed twice with 10 mM Tris-HCl (pH 8.0) and then collected by centrifugation (4,000 rpm, 15 min, 4°C) (Hitachi himac CR21G centrifuge) and resuspended in 2 mL of the same buffer. Cells were disrupted by sonication (10 pulses of 10 s, with intervals of 1 min on ice at 21% of amplitude) using the Sonics Vibra Cell equipment (High Intensity Ultrasonic Processor, 750 Watt model) with a 13 mm probe and centrifuged (13,000 rpm, 30 min, 4°C), and the supernatant (soluble proteins) was stored at −80°C.

**Immunoblot.** Anti-M. tuberculosis EPSPS (MtEPSPS) polyclonal antibody was produced immunizing a mouse with 50 mg of purified recombinant MtEPSPS containing Freund's incomplete adjuvant (Sigma-Aldrich, USA) (total volume of 100 $\mu$L) by subcutaneous route, followed by a booster injection after 1 month. The mouse was euthanized by deep isoflurane inhalation 1 month later, and blood was collected by descendant aorta. Serum was separated by centrifugation at 10,000 × g for 10 min, aliquoted, and stored at −80°C (44). Immunoblot was performed in triplicate. Approximately 30 $\mu$g of M. smegmatis total protein (crude extract) from detergent fraction was denatured at 70°C for 10 min, subjected to 12% sodium dodecyl sulfate-polyacrylamide gel electrophoresis (SDS-PAGE), and transferred to nitrocellulose membranes (Merck Millipore) in buffer Tris 25 mM, glycine 192 mM (pH 8.8), and methanol 20% for 4 h at 70 V. After transfer, the membrane was blocked with 5% nonfat dried milk (Santa Cruz Biotechnology) and 0.05% Tween 20 (Sigma-Aldrich) in Tris-buffered saline (T-TBS) (2 h, 4°C) and probed with anti-MtEPSPS polyclonal mouse antibody in a 1:500 dilution (overnight at 4°C). Membranes were washed three times with T-TBS, and alkaline phosphatase-conjugated anti-mouse secondary antibody (Invitrogen) was used at a dilution of 1:5,000 (45). Chemiluminescent substrate (Novex by Life Technologies, USA) was used for detection with ChemiDoc (Bio-Rad). Densitometric analysis was performed using the Image Lab version 6.1.0 Standard Edition software (Bio-Rad Laboratories, Inc.).

**Recombinant production of WT and mutant forms of MsEPSPs.** E. coli cells transformed with recombinant pET-23a(+) plasmids carrying a WT or mutated copy (R134A, E321N, or D61W) of the aroA gene from M. smegmatis were selected on solid LB with ampicillin. A single colony was grown in LB medium (5 mL) with antibiotic, at 37°C, overnight. Precultivated inocula were then diluted (1:1,000) in fresh LB (for WT, R134A, and E321N) or Terrific broth (TB) medium (for D61W) containing ampicillin. After reaching an OD$_{600}$ of 0.4 to 0.6, cultures were grown for 23 h at 37°C, under shaking (180 rpm) and aerobic conditions. Protein production was achieved without isopropyl $\beta$-D-1-thiogalactoside (IPTG) induction. Cells were harvested by centrifugation (11,800 × g for 30 min, at 4°C) and stored at −20°C. As a negative control of protein production, the same procedure was employed for E. coli cells carrying pET-23a(+) without the aroA gene (pET23a[+]::Ø). The production of soluble proteins was confirmed by 12% SDS-PAGE stained with Coomassie brilliant blue.

**Purification of recombinant proteins by liquid chromatography.** Recombinant WT and mutant proteins were purified using two or three chromatographic steps. All purification steps were carried out in an ÄKTA system (GE Healthcare Life Sciences) at 4°C. Approximately 3.2 g of cells overproducing each protein was collected by centrifugation for 20 min at 6,000 × g (4°C). Cells were suspended in 25 mL of 50 mM Tris-HCl

(pH 7.8; buffer A) and incubated for 30 min in the presence of 0.2 mg/mL lysozyme (Sigma-Aldrich) under slow stirring. Cells were disrupted by sonication (4 pulses of 20 s, with intervals of 1 min on ice, at 60% of amplitude). Cell debris were removed by centrifugation (11,800 $\times$ *g* for 60 min, at 4℃). The supernatant was incubated with 1% (wt/vol) of streptomycin sulfate (Sigma-Aldrich) for 30 min at 4℃, under gentle stirring, and centrifuged. The supernatant was dialyzed twice against 2 L of buffer A using a dialysis tubing with a cutoff filter of 12 to 14 kDa. The samples were clarified by centrifugation and loaded onto a Q-Sepharose Fast Flow (GE Healthcare Life Sciences) column, preequilibrated with buffer A. Adsorbed proteins were eluted by a 20-column volume (CV) linear gradient (0 to 100%) of 50 mM Tris-HCl, 1 M NaCl (pH 7.8; buffer B) at 1 mL/min flow rate. Protein elution was monitored by UV detection at 215, 254, and 280 nm. Eluted fractions containing the protein of interest were pooled and ammonium sulfate was added to a final concentration of 1 M. After an incubation period of 30 min at 4℃, and subsequent centrifugation, the supernatant was loaded onto a HiLoad 16/10 Phenyl Sepharose HP (GE Healthcare Life Sciences) column, preequilibrated with 50 mM Tris-HCl, 1 M $(NH_4)_2SO_4$ (pH 7.8; buffer C). Proteins were eluted by a 20 CV linear gradient (100 to 0% ammonium sulfate) in buffer A, at 1 mL/min flow rate. For WT and R134A mutant, a Mono Q HR 16/10 (GE Healthcare Life Sciences) column was used as a third step. Protein fractions eluted from the second column were pooled, centrifuged, and loaded onto the last column, preequilibrated with buffer A. Proteins were eluted by a 15 CV linear gradient in buffer B (0 to 100%) at 2 mL/min flow rate, pooled and dialyzed against buffer A, and finally stored at −80℃. All protein fractions were analyzed by 12% SDS-PAGE stained with Coomassie brilliant blue. Protein homogeneity above 95% was checked by densitometry using GelDoc (Bio-Rad) equipment. Protein concentration was determined by the bicinchoninic acid (BCA) method (Thermo Scientific Pierce BCA protein assay kit).

**Protein identification by LC-MS/MS.** Recombinant *Ms*EPSPS enzymes were precipitated with chloroform/methanol (46). Pellets were resuspended in 100 mM Tris-HCl buffer (pH 7.0) containing 8 M urea (Affymetrix USB), and disulfide bonds were reduced in 5 mM dithiothreitol (DTT) (Ludwig Biotec) for 20 min at 37℃. After that, cysteine residues were alkylated with 25 mM iodoacetamide (IAM) (Sigma-Aldrich) for 20 min at room temperature in the dark. Urea was diluted to 2 M with 100 mM Tris-HCl (pH 7.0) and trypsin (Promega) was added at a mass ratio of 1:100 (trypsin/protein). Protein digestion was performed overnight at 37℃. Formic acid (Merck) was added to quench the reaction (5% vol/vol, final concentration). Tryptic peptides were then separated in a reversed phase $C_{18}$ (5 $\mu$m ODS-AQ $C_{18}$, Yamamura Chemical Lab) column using a nanoUPLC (nanoLC Ultra 1D plus, Eksigent) and eluted (400 nL/min) with acetonitrile gradient (5% to 80%) (LiChrosolv, Merck) with 0.1% formic acid. Eluting peptide fragments were ionized by electrospray ionization and analyzed on an LTQ-XL Orbitrap Discovery hybrid instrument (Thermo Fisher Scientific). The LC-MS/MS procedure was performed according to the data-dependent acquisition (DDA) method. Precursors were collected from 400 to 1,600 *m/z* at 30,000 resolution in the Orbitrap, and the eight most abundant ions per scan were selected to collision-induced dissociation (CID), using helium as the collision gas in the ion trap. Raw files were searched in the PatternLab for Proteomics platform (47) with a database containing forward and reverse *E. coli* BL21-DE3 reference proteome and *Ms*EPSPS WT and mutant sequences using Comet (48). Carbamidomethyl was set as a fixed modification. Search results were filtered to a false discovery rate of 1% through the module Search Engine Processor from PatternLab for Proteomics.

**EPSPS enzyme activity assays.** Recombinant *Ms*EPSPS enzymes were assayed using a continuous spectrophotometric coupled assay (49). Enzyme activity was determined by measuring the kinetics of $P_i$ release from the reaction catalyzed by EPSPS in the forward direction: phosphoenolpyruvate (PEP) + 3-phosphoshikimate (S3P) $\rightleftharpoons$ inorganic phosphate ($P_i$) + 5-enolpyruvylshikimate-3-phosphate (EPSP). $P_i$ release was measured in a coupled assay with purine nucleoside phosphorylase from *M. tuberculosis* (*Mt*PNP; EC 2.4.2.1) and 2-amino-6-mercapto-7-methylpurine ribonucleoside (MESG) (49), which was synthesized according to a published protocol (50) (Fig. S2). All EPSPS activity assays were performed in 100 mM Tris-HCl buffer (pH 7.8), at 25℃ for 3 min, using 138 nM *Mt*PNP, 1.7 nM *Ms*EPSPS, and various concentrations of S3P and PEP. We adapted the previously published assay (51) and substituted Shikimate Kinase, ATP, and shikimate by a commercially available S3P. Apparent steady-state kinetic constants were determined by monitoring the WT and mutant EPSPS activities at varying concentrations of PEP (Sigma-Aldrich) and fixed-saturating concentrations of S3P (Sigma-Aldrich) (Table S1). All measurements were performed in a 1.0 cm path length quartz cuvette, in duplicate, and the rate of $P_i$ production was measured in a UV/Vis spectrophotometer (Shimadzu). Steady-state kinetic constants were obtained by nonlinear regression analysis of the kinetic data fitted to the Michaelis-Menten equation ($v = V_{max} \times [S]/[K_m + (S)]$) using the SigmaPlot 14.0 software (SPSS, Inc.). The $k_{cat}$ (catalytic constant) values were calculated using the corresponding equation ($k_{cat} = V_{max}/[E_t]$), where $V_{max}$ is the maximum velocity of the enzyme reaction and $E_t$ is the total enzyme concentration.

## SUPPLEMENTAL MATERIAL

Supplemental material is available online only.

**SUPPLEMENTAL FILE 1**, PDF file, 1 MB.

## ACKNOWLEDGMENTS

We thank Sara Fortune and Jeremy Rock for providing the pLJR962 plasmid and Hector Morbidoni and Luis Saraiva Timmers for insightful discussions. C.V.B., P.M., and L.A.B. would like to acknowledge financial support given by CNPq/FAPERGS/CAPES/ BNDES to the National Institute of Science and Technology on Tuberculosis (INCT-TB),

Brazil (grant numbers: 421703-2017-2/17-1265-8/14.2.0914.1). C.V.B. (310344/2016-6), P.M. (305203/2018-5), and L.A.B. (520182/99-5) are research career awardees of the National Council for Scientific and Technological Development of Brazil (CNPq). This study was financed in part by the Coordenação de Aperfeiçoamento de Pessoal de Nível Superior—Brasil (CAPES)—Finance Code 001.

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
