## [Reviewer comments · Microbiology Spectrum]

Microbiology Spectrum

EPSP synthase-depleted cells are aromatic amino acid auxotrophs in *Mycobacterium smegmatis*

Mario Duque-Villegas, Bruno Abbadi, Paulo Romero, Letícia Matter, Luiza Galina, Pedro Dalberto, Valnês Rodrigues-Junior, Rodrigo Ducati, Candida Roth, Raoní Rambo, Eduardo de Souza, Márcia Perello, Pablo Machado, Luiz Basso, and Cristiano Bizarro

Corresponding Author(s): Cristiano Bizarro, Pontifícia Universidade Católica do Rio Grande do Sul

Review Timeline:

Submission Date:	April 7, 2021
Editorial Decision:	May 1, 2021
Revision Received:	September 8, 2021
Editorial Decision:	September 27, 2021
Revision Received:	November 15, 2021
Accepted:	November 17, 2021

Editor: Amanda Oglesby

Reviewer(s): Disclosure of reviewer identity is with reference to reviewer comments included in decision letter(s). The following individuals involved in review of your submission have agreed to reveal their identity: Hyungjin Eoh (Reviewer #1)

Transaction Report:

DOI: <https://doi.org/10.1128/Spectrum.00009-21>

May 1, 2021

Prof. Cristiano Valim Bizarro
Pontifícia Universidade Católica do Rio Grande do Sul
Porto Alegre
Brazil

Re: Spectrum00009-21 (Evaluating *aroA* gene essentiality and EPSP synthase vulnerability in *Mycobacterium smegmatis* under different nutritional conditions)

Dear Prof. Cristiano Valim Bizarro:

Thank you for submitting your manuscript to Microbiology Spectrum. Your manuscript has been reviewed by three experts, who have made several suggestions for revision. In particular, reviewers 1 and 2 felt that the current studies do not sufficiently support one of the manuscript's conclusions - that AroA is validated as an *M. tuberculosis* drug target. I tend to agree with this concern. I would be happy to consider a revised manuscript that addresses this and other comments provided by the reviewers. When submitting the revised version of your paper, please provide (1) point-by-point responses to the issues raised by the reviewers as file type "Response to Reviewers," not in your cover letter, and (2) a PDF file that indicates the changes from the original submission (by highlighting or underlining the changes) as file type "Marked Up Manuscript - For Review Only". Please use this link to submit your revised manuscript - we strongly recommend that you submit your paper within the next 60 days or reach out to me. Detailed information on submitting your revised paper are below.

Link Not Available

Sincerely,

Amanda Oglesby

Journals Department
Reviewer comments:

Reviewer #1 (Comments for the Author):

The manuscript seeks to validate the *M. tb* shikimate pathway as a source of drug targets. The shikimate pathway is linked to many metabolic pathways required for mycobacterial growth. Authors characterized the functional essentiality and phenotypic vulnerability of AroA enzyme (6th step of the shikimate pathway) to address the question.

Some comments are followed:

1. Authors propose AroA as a potential drug target. Unfortunately, the essentiality of AroA is dependent upon the environmental nutrient condition. AroA deficient *M. smegmatis* failed to grow in 7H9 media but the growth was restored by supplementing with aromatic amino acids. Recent papers (PMID: 23911587, PMID: 31825837) suggested that *M. tuberculosis* at the site of infection shares nitrogen sources and amino acids from host macrophages. Thus, authors may include the viability test of *aroA* mutants under host cells.

2. It will be interesting to test if *aroA* knock-out strain can be obtained by plating LB or 7H9 with aromatic amino acids.
3. Fig. 2C. There is no strain containing other mutant *aroA* gene.
Fig. 2D. This reviewer can't find the sentence explaining Fig. 2D.
4. Fig. S1 legend. Correct all typos.
5. EPSPS enzyme assay coupling MESG to measure released Pi. The method section didn't explain the components in in vitro enzyme reaction. To monitor the released Pi, the reaction should contain ATP.
6. Table 3: Kcat/Km value has wrong unit ($\mu\text{M}^{-1}\text{s}^{-1}$)
7. Fig. 3. % mRNA suppression of *aroA* gene in PAM1, 2, and 3 should be provided.
mmpL3 was used as a positive control. mmpL3 CRISPRi protein vulnerability and mRNA suppression levels should be provided.
8. Fig S6 and S7: It should include WT lysate to compare the basal level of EPSPS.
9. Fig. 4B. PAM1, 2, and 3 were grown in 7H10 regardless of *aroA* suppression if the media contains aromatic amino acid. As shown in A, Mtb shikimate pathway is involved in many pathways including folates and menaquinone. Based upon the results of Fig. 4, can we conclude this pathway is exclusively involved in the biosynthesis of aromatic amid acid but the biosynthesis of other end products is shared by other metabolic activities ? Please discuss this in the discussion section.

Reviewer #2 (Comments for the Author):

The manuscript by Duque-Villegas and colleagues describes the characterization of *M. smegmatis* AroA. The authors use standard allelic exchange and Crispri to knockout/knockdown *aroA* in *M. smegmatis* and demonstrate phenotypes in defined 7H9 medium, but not LB. This was rescued upon the addition of aromatic amino acids. They investigate the kinetics of purified AroA and AroA point substituted proteins.

The study and results obtained are incremental. The author's premise is that these studies validate the development of drugs against Mtb that target AroA - though I would argue that this is a stretch with the current data set. Ideally such studies would be conducted in Mtb and would demonstrate essentially in vitro and in vivo using the Crispri knockdown strains induced at different stages of infection. Heterologous complementation of the *smegmatis* mutants with wild type and point substituted Mtb constructs would better support their model.

There are some issues with the presentation of the data that complicate interpretation.

Figure 1 depicting the Crispri knockdown should be supplemental.

The results text describing Figure 2 is confusing. It does not appear that there are any significant differences in the growth of strains in Fig 2C (stats?). Are these strains all merodiploid? There is no indication in the results for why the three amino acids were chosen for mutation. This info is found in the discussion. Clustal alignment in Figure 2D should have the entire protein and homology of proteins indicated in the results text.

The growth of Crispri strains in Figure 3 - the authors should indicate that mmpL3 is "essential" rather than "vulnerable" in the results text so that the reader understands how it is a suitable control.

Reviewer #3 (Comments for the Author):

Summary

This manuscript describes the establishment of the *aroA* gene of *M. smegmatis* (Msm) as essential. The motivation was to assess the enzymology and essentiality of this enzyme as a potential drug target for Mtb. Overall, the manuscript is very well written and the conclusions are reasonable and sound.

Specific comments:

The *aroA* deletion attempts convincingly support the conclusion that the AroA protein in its active forms is essential. In the later part of the paper, the authors used CRISPRi to further support this result. In the CRISPRi work the authors tested media

supplemented with aromatic amino acids and showed that the essentiality of AroA was lost, which makes sense. It was unclear to me that the authors tried making their gene deleted strains on this media, then tested it on regular media. I suspect they did but did not make it clear, or did not explain why it didn't work. Also, what was the recipe for this supplemented media? It was not indicated in the methods.

Lines 413-419: This paragraph and subtitle are somewhat confusing. Basically, the authors made a mutant with no phenotype, that's all that needs to be said. The subtitle sounds like they're making a super enzyme.

End of the Results section. The last paragraph basically has the nice result that they can functionally complement an aroA knockdown by supplementing the media...not sure why the authors found this "interesting" rather than "as expected".

Minor:

- mention of the figures started in the Methods, which isn't standard procedure. It would help for the authors to reference Fig 1 and after starting in the Results section, or at least mention them in the results section in order.
- the terms "knock out (KO)" and "knocked out" are jargon in bacteriology (knock down is ok for lack of a better term). More specific descriptors ("deletion", "deletion and disruption", "mutated" etc) would be more appropriate. As this is a nicely written manuscript, it would be that much better if proper genetic terminology were used. "CO" for complementation vectors also doesn't make sense to me.
- "western blot" is also jargon. A more appropriate term is "immunoblot"
- line 125: "the CRISPRi" should be "a CRISPRi"...there's more than one system to my knowledge.
- the authors should explain the use of the mmpL3 target in the Results and not in the discussion.
- line 299 subtitle doesn't make sense. Do you mean "overexpression of... genes"? Also, proteins are "produced" not "expressed"; genes are expressed (I realize this is a very common mistake).
- All growth curves should be presented in log scale. P values for Fig 2C are warranted.
- I believe almost all of the supplemental material can be incorporated into the main text (and maybe some could be deleted if not essential to show).

Staff Comments:

Preparing Revision Guidelines

For complete guidelines on revision requirements, please see the Instructions to Authors at [link to page]. **Submissions of a paper that does not conform to Microbiology Spectrum guidelines will delay acceptance of your manuscript.**

Due to the SARS-CoV-2 pandemic, our typical 60 day deadline for revisions will not be applied. I hope that you will be able to submit a revised manuscript soon, but want to reassure you that the journal will be flexible in terms of timing, particularly if experimental revisions are needed. When you are ready to resubmit, please know that our staff and Editors are working remotely and handling submissions without delay.

If you would like to submit an image for consideration as the Featured Image for an issue, please contact Spectrum staff.

Reviewer comments:

Reviewer #1 (Comments for the Author):

The manuscript seeks to validate the *M. tb* shikimate pathway as a source of drug targets. The shikimate pathway is linked to many metabolic pathways required for mycobacterial growth. Authors characterized the functional essentiality and phenotypic vulnerability of AroA enzyme (6th step of the shikimate pathway) to address the question.

Some comments are followed:

1. Authors propose AroA as a potential drug target. Unfortunately, the essentiality of AroA is dependent upon the environmental nutrient condition. AroA deficient *M. smegmatis* failed to grow in 7H9 media but the growth was restored by supplementing with aromatic amino acids. Recent papers (PMID: 23911587, PMID: 31825837) suggested that *M. tuberculosis* at the site of infection shares nitrogen sources and amino acids from host macrophages. Thus, authors may include the viability test of *aroA* mutants under host cells.

We agree with reviewer 1 that this is an important issue. Taken together our previous data and results from experiments conducted to answer reviewer's questions, we show that in defined growth medium *aroA*-deficient cells can be rescued by supplementation with aromatic amino acids (AroAAs) and also that the growth impairment of *aroA*-knockdown cells can be rescued with the same supplement (L-tryptophan, L-phenylalanine and L-tyrosine). Therefore, *aroA* is essential and EPSPS vulnerable only in growth conditions without or with insufficient availability of AroAAs. To make this clear, we introduced in our Discussion section the followings pieces of text:

Lines 528-533 (*aroA* essentiality):

*"Here, using *M. smegmatis* as a mycobacterial model organism, we show that the *aroA* gene, which codes for the enzyme 5-enolpyruvylshikimate-3-phosphate synthase (EPSPS) from the shikimate pathway, is essential only when sufficient amounts of L-tryptophan, L-phenylalanine and L-tyrosine (AroAAs) are not available (Fig. 1). We are not aware of previous attempts to generate *aroA*-deficient strains in *M. smegmatis*."*

Lines 567-586 (EPSPS vulnerability):

*"Next, we addressed the issue of target vulnerability. A target should not be only essential but also vulnerable, otherwise chances are low to develop bioactive compounds that effectively kill or impart growth defects on infective agents. To evaluate MsEPSPS vulnerability, we performed gene knockdown experiments using a CRISPRi system developed for mycobacteria (20). Using an in-house Python script, we selected target sequences adjacent to PAM motifs whose repression strengths were characterized previously (20). The experiments were conducted in both rich (LB, Fig. 2A-D) and defined (7H9 or 7H10, Fig. 2E-H) media, with markedly different results. As expected, the sgRNA control targeting the *mmpL3* gene was found to be vulnerable in both nutritional conditions, either in liquid or solid media. Silencing *mmpL3* caused a cessation of bacterial growth by 15 h, and using the drop method on plates, we have observed a reduction of at least 1,000-fold in the CFU counting (Fig. 2A and 2E). Differently from *MmpL3*, MsEPSPS was found to be vulnerable only in defined medium, irrespective of PAM's repression strength. Silencing *aroA* gene caused an impairment of the bacterial growth after 18 h, but not a complete cessation (Fig. 2F-H). This suggests that the abrupt reduction on endogenous MsEPSPS levels observed*

for *aroA*-knockdown cells (Fig. 3) does not cause bacterial death, but rather, growth impairment. Moreover, supplementation with AroAAs is sufficient to rescue the growth impairment of *aroA*-knockdown strains (Fig. 4B-D).”

In our view, however, it is still an open question whether the host microenvironment experienced by Mtb cells infecting a TB patient would supply the necessary amounts of AroAAs to eventually bypass the inhibition of EPSPS activity. The references provided by reviewer 1 (PMID: 23911587 and PMID: 31825837) are two seminal works from McFadden’s group in which sophisticated techniques of metabolic flux analyses were employed to elucidate the non-steady state metabolism of carbon (PMID: 23911587) and nitrogen (PMID: 31825837) in intracellular infecting *M.tuberculosis* (Mtb) cells. Indeed, as indicated by Reviewer 1, many amino acids were found to be obtained by infecting bacilli directly from the host cell, including alanine, glutamate/glutamine, aspartate/asparagine, valine, leucine and glycine, with one or both approaches (carbon and nitrogen tracing). However, both studies do not indicate that AroAAs can be obtained from the host cells in amounts to suffice the metabolic requirements of Mtb bacilli. On the opposite, in the nitrogen flux analysis study, the authors show that the amino acids phenylalanine and tyrosine, among others, need to be synthesized de novo to suffice biomass requirements, while the levels of tryptophan were too low to perform the analysis (PMID: 31825837). Moreover, an auxotrophic mutant strain of *M. tuberculosis* for tryptophan (defined mutation on *trpD* gene from the tryptophan biosynthesis pathway) was severely attenuated, with more than 90% of bacterial cells killed after 10 days of infection of bone marrow-derived macrophages. This same strain was completely avirulent after mice infection. (PMID: 11160012), indicating that at least tryptophan is not available in sufficient amounts for Mtb bacilli under these conditions.

Lastly, reviewer 1 suggested including a viability test of *aroA* mutants under host cells. Let me first point out some facts to frame our view about this suggestion. *M. smegmatis* is a fast growing saprophytic mycobacterial species traditionally used as a model organism in genetic studies of mycobacteria. It is easy to manipulate in the laboratory, without the biosafety concerns of Mtb cells, with colonies generating after plating within 2-3 days (PMID: 20036184). As any model organism, however, it has some limitations. Pathogenic mycobacteria, such as *M. tuberculosis*, survive within macrophages by preventing phagosomal maturation into an active lysosomal compartment (PMID: 17850480). In contrast, *M. smegmatis* is an avirulent species that is rapidly killed within macrophages. In this context, *M. smegmatis* has been used as useful model to study the killing mechanisms inside macrophages (PMID: 16681836) and also to identify mycobacterial genes that could afford some limited capacity to survive in this environment (PMID: 22363734). However, as this organism possess a different survival kinetics within macrophages when compared to Mtb cells, we don’t think it would be an appropriate model to study the effect of EPSPS depletion (by gene knockout or knockdown) in the intracellular environment.

2. It will be interesting to test if *aroA* knock-out strain can be obtained by plating LB or 7H9 with aromatic amino acids.

As described in the revised version of this manuscript, we have performed the *aroA* gene knockout experiment in defined 7H9/7H10 medium supplemented with AroAAs. We would like to thank reviewer 1 for this suggestion as we think the results obtained greatly improved the quality of our work. We were able to obtain viable colonies of *aroA*-deleted cells grown in media supplemented with AroAAs. As we have also shown that *aroA* is essential when selection is performed in LB or 7H9/7H10 medium without AroAA supplementation, our data demonstrates that *aroA*-deleted *M.smegmatis* cells are auxotrophic for AroAAs. In light of that, we performed significant changes in our manuscript, particularly in the Introduction, Results and Discussion sections. As we found AroAAs are sufficient to revert the effects of both *aroA* deletion (in terms of viability) and *aroA* knockdown (in terms of growth), we reorganized the presentation of our data, putting these two sets of experiments together and separating them from the experiments related to the evaluation of essentiality of particular residues and enzyme kinetics. We expect that in this way the message of our work is now much clearer.

3. Fig. 2C. There is no strain containing other mutant *aroA* gene.

We were only able to obtain viable *aroA*-deleted cells from the merodiploid strains containing an extra copy of either the WT *aroA* gene or a mutant version of *aroA* gene coding for the D61W EPSPS mutant. Therefore, we concluded that residues R134 and E321 are essential. That is the reason why we performed the growth curve of only these two knockout strains (*aroA*_WT and *aroA*_D61W), together with controls, as presented in figure 2C.

3.1 Fig. 2D. This reviewer can't find the sentence explaining Fig. 2D.

Thank you for the observation. The previous Fig. 2D is now 5D in this revised version of the manuscript. We added a reference to fig. 5D on lines 499-500.

4. Fig. S1 legend. Correct all typos.

The figure legend was changed as indicated.

5. EPSPS enzyme assay coupling MESG to measure released Pi. The method section didn't explain the components in in vitro enzyme reaction. To monitor the released Pi, the reaction should contain ATP.

We measured the *Mt*EPSPS enzyme activity by monitoring the kinetics of Pi release from the reaction catalyzed by this enzyme: phosphoenolpyruvate (PEP) + 3-phosphoshikimate (S3P) \rightleftharpoons phosphate (Pi) + 5-enolpyruvylshikimate-3-phosphate (EPSP). To do so, we used a continuous spectrophotometric coupled assay with the purine nucleoside phosphorylase enzyme (PNP) and a guanosine analogue, 2-amino-6-mercapto-7-methylpurine ribonucleoside (MESG) (PMID: 1534409). In this original publication, the suitability of the assay was determined by monitoring the Pi released from ATP in two biochemical systems: the D-glyceraldehyde/glycerol kinase system and the actomyosin subfragment 1 ATPase system. Some of us have previously used the MESG coupled assay to monitor the EPSPS activity from *Mycobacterium tuberculosis* (PMID: 12699693). In this study, *Mt*EPSPS activity was coupled with the activity of the preceding enzyme of the Shikimate pathway, Shikimate Kinase (SK), which catalyzes the

following reaction: shikimate + ATP \rightleftharpoons 3-phosphoshikimate (S3P) + ADP. SK enzyme, shikimate and ATP were added to supply *Mt*EPSPS with one of its substrates, S3P. We adapted this protocol and supplied *Ms*EPSPS with a commercial source of S3P, and in this way we avoided the addition of the SK enzyme and the substrates shikimate and ATP in the reaction.

To make this clearer in the main text, we replaced the previous text to the following lines (395 – 497):

“EPSPS enzyme activity assays. Recombinant *Ms*EPSPS enzymes were assayed using a continuous spectrophotometric coupled assay (36). Enzyme activity was determined by measuring the kinetics of inorganic phosphate (Pi) release from the reaction catalyzed by EPSPS in the forward direction: phosphoenolpyruvate (PEP) + 3-phosphoshikimate (S3P) \rightleftharpoons inorganic phosphate (Pi) + 5-enolpyruvylshikimate-3-phosphate (EPSP). Pi release was measured in a coupled assay with purine nucleoside phosphorylase from *M. tuberculosis* (*Mt*PNP; EC 2.4.2.1) and 2-amino-6-mercapto-7-methylpurine ribonucleoside (MESG) (36), which was synthesized according to a published protocol (37) (Supplemental Material - Fig. S2). All EPSPS activity assays were performed in 100 mM Tris-HCl buffer pH 7.8, at 25°C for 3 min, using 138 nM of *Mt*PNP, 1.7 nM of *Ms*EPSPS and varying concentrations of S3P and PEP. We adapted the previously published assay (38) and substituted Shikimate Kinase, ATP and shikimate by a commercially available S3P.”

6. Table 3: Kcat/Km value has wrong unit ($\mu\text{M}^{-1}/\text{s}^{-1}$)

In our view, there is no fundamental difference in using $\text{M}^{-1}\text{s}^{-1}$ or $\mu\text{M}^{-1}\text{s}^{-1}$ (assuming reviewer 1 meant $\mu\text{M}^{-1}\text{s}^{-1}$ and not $\mu\text{M}^{-1}/\text{s}^{-1}$). We stick to the former because it complies with recommendations of the Nomenclature Committee of the International Union of Biochemistry (NC-IUB) for units of second-order rate constants (PMID: 6615450) and is widely used by the community working with chemical and enzyme kinetics (e.g., Raymond Chang, Physical Chemistry for the Chemical and Biological Sciences (ISBN: 1-891389-06-8); Robert Copeland, Evaluation of Enzyme Inhibitors in Drug Discovery (ISBN: 0-471-68696-4). The use of $\text{M}^{-1}\text{s}^{-1}$ is also in line with our published works where $k_{\text{cat}}/k_{\text{M}}$ data were presented (e.g., PMID: 28754992, PMID: 24407036, PMID: 23988349, PMID: 23671579, PMID: 23424660, among others).

7. Fig. 3. % mRNA suppression of *aroA* gene in PAM1, 2, and 3 should be provided. *mmpL3* was used as a positive control. *mmpL3* CRISPRi protein vulnerability and mRNA suppression levels should be provided.

Instead of measuring mRNA levels, we provided information on *Ms*EPSPS endogenous levels using the immunoblot technique. To do that, we prepared mouse polyclonal antibody against recombinant *M. tuberculosis* EPSPS (*Mt*EPSPS). Our results clearly indicate that endogenous *Ms*EPSPS is depleted in *aroA*-knockdown cells using anyone of the three different sgRNAs targeting PAM1, PAM2 or PAM2-adjacent sequences (Figs. S6 and S7). In our view, this is a much better experimental approach to evaluate the effectiveness of the knockdown process than measuring mRNA levels, as we confirm the result of the perturbation directly in terms of alterations in the cellular levels of the encoded protein, which is our purpose using this technique.

Concerning *mmpL3* experiments, we used *mmpL3*-knockdown cells solely as positive controls, to confirm that the technique is working properly. *MmpL3* protein was previously validated as a drug target using CRISPRi in mycobacteria (PMID: 31160289). The dependency on ATc concentration of both *mmpL3* mRNA relative levels and bacterial growth was reported, as well as increased sensitivity to *MmpL3* inhibitors in *mmpL3*-knockdown cells (same study). The *mmpL3* knockdown was recently used as a

positive control experiment in the same way we performed, monitoring the effect in terms of growth perturbation (PMID: 32423951). Taken our data together, the depletion in *MtEPSPS* levels directly observed by immunoblot only after ATc addition (with three different sgRNA constructs) and the growth perturbation observed (as expected) in *mmpL3*-knockdown cells strongly support that the knockdown technique is working in our hands.

8. Fig S6 and S7: It should include WT lysate to compare the basal level of EPSPS.

All the experiments performed and reported in the main text regarding growth perturbation of *aroA*-knockdown cells under different conditions involve comparisons of ATc-induced (+ATc) x ATc-uninduced (-ATc) cells. This makes sense because whatsoever alteration (e.g., metabolism, gene expression) inadvertently introduced by transforming cells with sgRNA constructs will also be present in our control (-ATc) cells. That is why we choose in immunoblot experiments -ATc cells as our reference for the basal level of endogenous *MtEPSPS* to compare with *aroA*-knockdown (+ATc) cells.

9. Fig. 4B. PAM1, 2, and 3 were grown in 7H10 regardless of *aroA* suppression if the media contains aromatic amino acid. As shown in A, *Mtb* shikimate pathway is involved in many pathways including folates and menaquinone. Based upon the results of Fig. 4, can we conclude this pathway is exclusively involved in the biosynthesis of aromatic amid acid but the biosynthesis of other end products is shared by other metabolic activities? Please discuss this in the discussion section.

Reviewer 1 raised here a very important issue. Our results strongly suggest that both *aroA*-deficient *M. smegmatis* cells are auxotrophic for aromatic amino acids (AroAAs) (knockout experiments) and that the growth impairment of *aroA*-knockdown cells can be completely rescued by supplementing the culture with AroAAs. Considering the pivotal role of the shikimate pathway of supplying cells with chorismate, this can be considered a surprising result. As depicted in our Fig. 4A, apart from the pathways leading to phenylalanine, tyrosine and tryptophan, chorismate is the starting compound for the p-aminobenzoate branch of folates, for the synthesis of ubiquinones from p-hydroxybenzoate, and for the synthesis of isochorismate, leading to naphthoquinones, menaquinones and mycobactins. Interestingly, in *M. tuberculosis*, the gene *aroK*, encoding Shikimate Kinase (*MtSK*), was found to be essential. *MtSK* precedes *MtEPSPS* in the shikimate pathway. The authors were not able to retrieve viable cells even after supplementing growth cultures with an “aromix” containing AroAAs, p-hydroxybenzoate, p-aminobenzoic acid and 2,3-dihydroxybenzoate (PMID: 12368440). This is in stark contrast with our results in *M. smegmatis*, but it should be noted it is more an exception than the rule when we broaden our analysis and consider the nutritional requirements of knockout strains from other bacterial species. For example, *aroA*-knockout strains from *Salmonella Infantis* (PMID: 31240343), *Pseudomonas aeruginosa* (PMID: 11854239) and *Burkholderia glumae* (PMID: 24754446) were found to be auxotrophic for the three AroAAs (tryptophan, phenylalanine and tyrosine): the supplementation with them was sufficient to rescue viable cells and/or growth, as we have found for *M. smegmatis*. By contrast, in *Aeromonas hydrophila* (PMID: 9573055), *Shigella flexneri* (PMID: 7927802), *Aeromonas salmonicida* (PMID: 8478107) and *Salmonella typhimurium* (PMID: 7015147), viable *aroA*-knockout cells were rescued after supplementing growth cultures with the three AroAAs and p-aminobenzoic acid (pAB). In the latter case, the additional supplementation with 2,3-dihydroxybenzoate (DHB), precursor of the enterochelin

siderophore, was made dispensable by the inclusion of ferrous sulphate on the growth medium (PMID: 7015147). Some variations on the “aromix” composition were employed to rescue viable *aroA*-knockout cells from other bacterial species: in *Pasteurella multocida*, it contained AroAAs, DHB and p-hydroxybenzoic acid (PMID: 1474900) and in *Bordetella pertussis*, the aromix contained AroAAs, DHB and pAB (PMID: 2407655). Noteworthy, most of these studies do not report attempts to rescue the cells with different supplement compositions, so we can not be certain that all the supplement components are absolutely required in each case.

Although the shikimate pathway is generally considered essential for bacteria, plants and fungi, the preceding data indicate there are still unsolved issues and discrepancies when we consider its essentiality in terms of metabolic requirements from downstream pathways. The discovery that mutations on genes encoding different enzymes of the same shikimate pathway can lead to different effects on growth requirement is also puzzling. In *Listeria monocytogenes*, mutations in *aroA* (which encodes the enzyme responsible for the first step of the pathway in this organism, DAHP synthase), *aroB* (encoding the second step enzyme, DHQ synthase), *aroE* (corresponding to *aroA* in other species, encoding EPSPS), and double mutants *aroA/B* and *aroA/E* have markedly differences in terms of growth rates in culture media, in intracellular growth behavior and in virulence attenuation (PMID: 15385459). The reasons for that are still not clear, but the authors speculate about the possibility of a low-level generation of the end product in some mutants and not in others, which could supply the metabolic requirements for some downstream pathways, but not for the more metabolically demanding ones, like the pathways leading to AroAAs synthesis. In the same vein, a mutation in the same gene on different organisms could conceivably lead to different growth requirement outcomes, depending on metabolic particularities of each organism.

Reviewer #2 (Comments for the Author):

The manuscript by Duque-Villegas and colleagues describes the characterization of M. smegmatis AroA. The authors use standard allelic exchange and Crispri to knockout/knockdown aroA in M. smegmatis and demonstrate phenotypes in defined 7H9 medium, but not LB. This was rescued upon the addition of aromatic amino acids. They investigate the kinetics of purified AroA and AroA point substituted proteins.

1. The study and results obtained are incremental. The author's premise is that these studies validate the development of drugs against Mtb that target AroA - though I would argue that this is a stretch with the current data set. (This cannot be concluded) Ideally such studies would be conducted in Mtb and would demonstrate essentially in vitro and in vivo using the Crispri knockdown strains induced at different stages of infection. Heterologous complementation of the smegmatis mutants with wild type and point substituted Mtb constructs would better support their model.

In light of Reviewer's 2 general perception of our manuscript, we critically reevaluated our work. We realized the main points of our work were not clear enough and some experiments seemed somewhat disconnected from others. Therefore, we performed an extensive revision of our text, as provided in this second version of the manuscript. We hope that the addition of novel data from knockout experiments, indicating that *aroA*-deleted cells are auxotrophic for AroAAs, together with the restructuring of our text, have greatly improved the quality of the manuscript. We show that the supplementation with only the three AroAAs (L-tryptophan, L-phenylalanine and L-tyrosine) was sufficient to both rescue viable *aroA*-deficient cells and growth impairment in *aroA*-knockdown cells. This is by no means an incremental result. The current view, based on knockout studies on the SK-encoding *aroK* gene from *M. tuberculosis* (PMID: 12368440), is that the essentiality of the mycobacterial shikimate pathway cannot be bypassed by supplementation with aromatic compounds, such as AroAAs, p-hydroxybenzoate, p-amino-benzoic acid and 2,3-dihydroxybenzoate. We discuss this contrasting results in the context of findings in other bacterial species, which we believe will make our work also an interesting reading to people working with metabolism in other bacteria. It is important to note that our claims are backed up by robust data: we performed all the steps to select double crossover events independently in 3-5 biological replicates for each nutritional condition evaluated (LB, 7H10, 7H10+AroAAs). The CRISPRi knockdown data was obtained using three different PAM-adjacent sequences, using PAMs of different strengths. Moreover, we are not aware of previous works on mycobacterial EPSPSs in which the essentiality of specific residues was evaluated. Our results of *aroA* knockout in merodiploid cells containing an extracopy of *aroA* encoding EPSPS point mutants were discussed and evaluated taking into account the functional impact of those mutations in recombinant forms produced and characterized in this same study, and not by inferences from data obtained elsewhere from enzyme's orthologs. This is by no means a common practice, probably because it is technically demanding to produce, purify and extract kinetic parameters from recombinant WT and mutant enzymes. In our view, the observation that catalytic residues are indeed essential strengthen the view that *aroA* gene essentiality (in the absence of AroAAs) depends on the EPSPS enzyme activity of *aroA*-encoded protein, which is usually taken for granted without further validation. In our opinion, the experimental approach employed, complementing data from knockout, knockdown, point mutant strains and kinetic analysis of recombinant proteins, is innovative and brought new insights about *aroA* gene essentiality and EPSPS vulnerability under different nutritional conditions.

Concerning Reviewer's 2 reticence about using *M. smegmatis* as a model system, it should be noted that this organism is an accepted research model widely used by the community that works with mycobacteria. Robust validation experiments with complementary tools can be performed in relatively shorter times, with lower cost and higher throughput. In our view, however, *M. smegmatis* is not a good model for *M. tuberculosis* to evaluate gene essentiality and target vulnerability inside host cells or in in vivo models. Our view about this issue is detailed in the response to question 1 of Reviewer 1. Further characterization of *aroA* gene essentiality and EPSPS vulnerability in experimental systems that more closely resembles the microenvironment experienced by infecting bacilli should be pursued directly with *Mtb* cells. We envision this as the natural next step of our work, together with the use of *aroA*-knockdown *M.smeg* cells generated as hypersensitive strains for compounds acting on EPSPS. These

strains can be used in whole cell target-based screening efforts to identify new hit compounds that inhibit EPSPS from chemical libraries.

Lastly, Reviewer 2 suggested using heterologous complementation of smegmatis mutants with wild type and point substituted Mtb constructs. While heterologous complementation is a useful technique to identify functionally related sequences from a different organism or to study functional equivalence in evolutionary studies, it is not clear to us what kind of additional information we could obtain with that approach in the context of our work. The effects of perturbations like gene knockout or gene knockdown in terms of cell viability or growth behavior under different nutritional conditions are expected to be consequences of alterations in the shikimate pathway as whole, with alterations on the production rates of the end product chorismate leading to perturbations in the metabolic fluxes of downstream pathways.

There are some issues with the presentation of the data that complicate interpretation.

2. Figure 1 depicting the Crispr knockdown should be supplemental.

We transferred Fig. 1 to Supplemental Material as Fig. S1.

3. The results text describing Figure 2 is confusing. It does not appear that there are any significant differences in the growth of strains in Fig 2C (stats?). Are these strains all merodiploid? There is no indication in the results for why the three amino acids were chosen for mutation. This info is found in the discussion. Clustal alignment in Figure 2D should have the entire protein and homology of proteins indicated in the results text.

3.1 The results text describing Figure 2 is confusing. It does not appear that there are any significant differences in the growth of strains in Fig 2C (stats?).

We have reorganized the structure of our figures to cope with the changes in our text. Figure 2C from the first version of our manuscript is Figure 5D in this second version. The data in this figure is from biological triplicates. As indicated by reviewer 3, we have plotted the OD readings in log scale. The growth differences between our control strains containing an empty copy of the vector pNIP40 integrated (pNIP40::∅) or the one with the original *aroA* deleted but containing a second copy of WT *aroA* gene integrated (*aroA*_WT) and the strain *aroA*_D61W were not statistically significant. We supplied the p-values on Figure 5D legend. We decided to suppress the data from the growth curve of the reference strain *M. smegmatis* mc²155 because we realized this is not a suitable control to compare with *aroA*-disrupted strains. This strain have not passed by the same experimental procedures and do not have the pNIP40/b vector integrated.

3.2 Are these strains all merodiploid?

The strains grown are not merodiploids. pNIP40::Ø is the reference *M. smegmatis* mc²155 strain containing plasmid pNIP40/b integrated. The latter is a plasmid vector derived from the temperate mycobacteriophage Ms6 that integrates into a specific chromosomal region (overlapping the 3' end of the tRNA^{Ala} gene) of *M. smegmatis*, *M. vaccae*, *M. bovis* BCG and *M. tuberculosis* (PMID: 9884232). *aroA*_WT and *aroA*_D61W strains are derived from merodiploid strains that contains an extracopy of WT or mutant D61W *aroA* gene, respectively, integrated into the Ms6 chromosomal integration site. Both strains (*aroA*_WT and *aroA*_D61W) have had their original *aroA* allele deleted by allele-exchange mutagenesis using the pPR27XylE vector and therefore are not merodiploid strains anymore.

3.3. There is no indication in the results for why the three amino acids were chosen for mutation. This info is found in the discussion.

We have included the following piece of text in the results section that deals with the selection of residues and the identity and similarity of EPSPS orthologues (lines 464-479):

Identification of essential amino acid residues in MsEPSPS. To evaluate the essentiality of specific residues from MsEPSPS, we constructed a set of merodiploid strains carrying an extra copy of a mutated *aroA* allele encoding D61W, E321N or R134A MsEPSPS mutants. To confirm the correspondence among residues from different enzymes, we aligned the EPSPS sequences from *E. coli*, *M. smegmatis* and *M. tuberculosis* (Fig. 5A). The sequences from *M. smegmatis* and *M. tuberculosis* are 68% identical and 78% similar, while *M. smegmatis* and *E. coli* EPSPS proteins are 31% identical and 52% similar. We selected residues from MsEPSPS located at the same positions of residues found to influence enzyme activity in *E. coli* and *M. tuberculosis* EPSPS orthologues. We found that the D61 residue from MsEPSPS corresponds to the D49 and D54 residues from EcEPSPS and MtEPSPS sequences, respectively. A D49A substitution in EcEPSPS lead to a 24,000 reduction in the enzyme's specific activity (21), while a D54A or D54W substitution in MtEPSPS was predicted to impact protein stability (22). We also found that two residues involved in the catalytic reaction of EcEPSPS (21), R124 and D313, corresponds to MsEPSPS R134 and E321, respectively (Fig. 5A).

3.4 Clustal alignment in Figure 2D should have the entire protein and homology of proteins indicated in the results text.

As indicated, we have replaced our previous Figure 2D by a new figure (Figure 5A) displaying the entire alignment.

4. The growth of Crispr strains in Figure 3 - the authors should indicate that *mmpL3* is "essential" rather than "vulnerable" in the results text so that the reader understands how it is a suitable control.

As indicated, we included the following piece of text in the results section (lines 440-448):

"The vulnerability of this gene was evaluated in both rich media (solid and liquid LB - Fig. 2A-D) and defined media (solid 7H10 and liquid 7H9 - Fig. 2E-H) in the presence or absence of ATc 100 ng/mL, using the gene encoding the vulnerable MmpL3 protein (the essential mmpL3 gene) as a positive control. This gene codes for the mycobacterial membrane MmpL3 protein, which is responsible for trehalose monomycolate transport through the cell inner membrane (39). In M. tuberculosis and M. smegmatis, it was shown that silencing mmpL3 expression disrupts bacterial growth (20,40), leading to accumulation of TDM and cell death (41)."

Reviewer #3 (Comments for the Author):

Summary

This manuscript describes the establishment of the *aroA* gene of *M. smegmatis* (Msm) as essential. The motivation was to assess the enzymology and essentiality of this enzyme as a potential drug target for Mtb. Overall, the manuscript is very well written and the conclusions are reasonable and sound.

Specific comments:

1. The *aroA* deletion attempts convincingly support the conclusion that the AroA protein in its active forms is essential. In the later part of the paper, the authors used CRISPRi to further support this result. In the CRISPRi work the authors tested media supplemented with aromatic amino acids and showed that the essentiality of AroA was lost, which makes sense. It was unclear to me that the authors tried making their gene deleted strains on this media, then tested it on regular media. I suspect they did but did not make it clear, or did not explain why it didn't work. Also, what was the recipe for this supplemented media? It was not indicated in the methods.

We decided to evaluate whether *aroA* gene essentiality depends on the nutritional context by performing gene disruption experiments in both LB (which is a rich medium) and 7H10 (a defined “poor” medium). As suggested by reviewer 1 (question 2), we included in this revised version of our manuscript data from a third knockout experiment, in which we successfully selected viable *aroA*-deleted cells by using 7H10 medium supplemented with the aromatic amino acids L-tryptophan, L-phenylalanine and L-tyrosine (AroAAs). In this way, we show that *aroA*-deleted *M. smegmatis* cells are auxotrophic for AroAAs. By supplemented media, we mean the addition of this three AroAAs. We hope that we have made that clearer in this revised version of our manuscript.

2. Lines 413-419: This paragraph and subtitle are somewhat confusing. Basically, the authors made a mutant with no phenotype, that's all that needs to be said. The subtitle sounds like they're making a super enzyme.

Indeed. We would like to thank Reviewer 3 for this observation. We have replaced the previous text with a new paragraph:

Previous text (lines 413-419 in the first version of the manuscript):

“Mutations in the Asp61 residue of MsePSPS enzyme enables mycobacterial growth. Growth curves were performed to evaluate the impact of aspartic acid 61 residue (D61W) mutation of MsePSPS, on bacilli grown, in LB medium. We found no differences in the D61 mutant growth, when compared to control strains (Fig. 2C). This suggests that the replacement of this hydrophilic amino acid by the hydrophobic tryptophan residues does not abolish the MsePSPS activity inside cells.”

New paragraph (lines 493-500 of the revised version of the manuscript):

“Substitution of MsEPSPS aspartate 61 by tryptophan (D61W) does not impact *M. smegmatis* growth. Next, we evaluated whether the D61W mutation on MsEPSPS impacts bacilli growth. Growth curves were performed in LB medium for the *aroA*-deleted strain derived from the D61W merodiploid strain, together with controls: *aroA*-deleted strain derived from the WT merodiploid strain (extra copy of WT *aroA* gene), *M. smegmatis* mc²155 wild-type strain and pNIP40::∅ - integrated strain. We found that D61W mutation on MsEPSPS does not impact bacilli growth (Fig. 5D).”

3. End of the Results section. The last paragraph basically has the nice result that they can functionally complement an *aroA* knockdown by supplementing the media...not sure why the authors found this "interesting" rather than "as expected".

In this revised version of the manuscript, we tried to make the relevance of our findings clearer by putting our results into perspective with previous works not only in mycobacteria but also in other bacterial species. In particular, the abilities to both retrieve viable *aroA*-deleted cells and to rescue growth impairment from *aroA*-knockdown cells by supplementing growth media with AroAAs are treated in lines 528-566:

*“Here, using *M. smegmatis* as a mycobacterial model organism, we show that the *aroA* gene, which codes for the enzyme 5-enolpyruvylshikimate-3-phosphate synthase (EPSPS) from the shikimate pathway, is essential only when sufficient amounts of L-tryptophan, L-phenylalanine and L-tyrosine (AroAAs) are not available (Fig. 1). We are not aware of previous attempts to generate *aroA*-deficient strains in *M. smegmatis*. Considering the pivotal role of the shikimate pathway of supplying cells with the end-product chorismate, this can be considered a surprising result. Apart from the pathways leading to L-tryptophan, L-phenylalanine and L-tyrosine, chorismate is the starting compound for the p-aminobenzoate branch of folates, for the synthesis of ubiquinones from p-hydroxybenzoate, and for the synthesis of isochorismate, leading to naphthoquinones, menaquinones and mycobactins (Fig. 4A).*

*The entire shikimate pathway was deemed essential in the related *M. tuberculosis* (14). This conclusion was based on the inability to retrieve viable *M. tuberculosis* cells deficient in *aroK*, the gene that codes for the preceding enzyme in this pathway, Shikimate Kinase. Interestingly, viable cells could not be retrieved even with the addition of a supplement containing AroAAs, p-hydroxybenzoate, p-amino-benzoic acid and 2,3-dihydroxybenzoate (14). These metabolites can be synthesized in mycobacteria from the end product of the shikimate pathway, chorismate, and are part of known metabolic pathways that use chorismate as a starting compound. Noteworthy, in the absence of redundant activities, *aroK* and *aroA* deletion are expected to have the same impact in terms of chorismate production.*

*Nevertheless, our findings are not an exception, if we compare with similar studies conducted with other bacteria. For example, *aroA*-deleted strains from *Salmonella infantis* (17), *Pseudomonas aeruginosa* (18) and *Burkholderia glumae* (19) were found to be auxotrophic for the three AroAAs (L-tryptophan, L-phenylalanine, and L-tyrosine). For other bacterial species, the supplementation with aromatic compounds was also sufficient to rescue viable *aroA*-deleted cells, but the “aromix” included additional components other than AroAAs. For instance, in *Aeromonas hydrophila* (42), *Aeromonas salmonicida* (43), *Shigella flexneri* (44), and *Salmonella typhimurium* (45), viable *aroA*-deleted cells were rescued after supplementing growth cultures with the three AroAAs and p-aminobenzoic acid (pAB). Some variations on the “aromix” composition were employed to rescue viable *aroA*-deleted cells from other bacterial species. In *Pasteurella multocida*, it contained AroAAs, DHB and p-hydroxybenzoic acid (46) and in *Bordetella pertussis*, the aromix contained AroAAs, DHB and pAB (47). Although the shikimate pathway is generally considered essential for bacteria, plants and fungi, clearly there are still unsolved issues and discrepancies when we consider its essentiality in terms of metabolic requirements from downstream pathways.”*

Minor:

4. mention of the figures started in the Methods, which isn't standard procedure. It would help for the authors to reference Fig 1 and after starting in the Results section, or at least mention them in the results section in order.

We would like to thank reviewer 3 for this observation. We have reorganized our figures from both the main text and supplemental material and in this revised version of our manuscript the figures presented in the main text are mentioned in order from the results section.

5. the terms "knock out (KO)" and "knocked out" are jargon in bacteriology (knock down is ok for lack of a better term). More specific descriptors ("deletion", "deletion and disruption", "mutated" etc) would be more appropriate. As this is a nicely written manuscript, it would be that much better if proper genetic terminology were used. "CO" for complementation vectors also doesn't make sense to me.

We thank Reviewer 3 for helping us to improve the quality of our text. We replaced all instances containing Knockout, knock out and knocked out by more precise terms. The terms CO and complementation were completely suppressed because in fact we have not complemented mutant strains; we have mutated merodiploid strains.

6. "western blot" is also jargon. A more appropriate term is "immunoblot"

Accordingly, we have replaced all references to western blot by immunoblot.

7. line 125: "the CRISPRi" should be "a CRISPRi"...there's more than one system to my knowledge.

We have made the change as indicated.

8. the authors should explain the use of the mmpL3 target in the Results and not in the discussion.

As indicated, we have transferred the explanation about mmpL3 from the discussion to the results section (lines 440-448):

"The vulnerability of this gene was evaluated in both rich media (solid and liquid LB - Fig. 2A-D) and defined media (solid 7H10 and liquid 7H9 - Fig. 2E-H) in the presence or absence of ATc 100 ng/mL, using the gene encoding the vulnerable MmpL3 protein (the essential mmpL3 gene) as a positive control. This gene codes for the mycobacterial membrane MmpL3 protein, which is responsible for trehalose monomycolate transport through the cell inner membrane (39). In M. tuberculosis and M. smegmatis, it was shown that silencing mmpL3 expression disrupts bacterial growth (20,40), leading to accumulation of TDM and cell death (41)."

9. line 299 subtitle doesn't make sense. Do you mean "overexpression of... genes"? Also, proteins are "produced" not "expressed"; genes are expressed (I realize this is a very common mistake).

Thank you for this observation. We have revised and replaced all references to “protein expression” by “protein production”.

10. Change title

As indicated, we changed the title of our work:

Previous title:

*“Evaluating *aroA* gene essentiality and EPSP synthase vulnerability in *Mycobacterium smegmatis* under different nutritional conditions”*

Modified title:

*“EPSP synthase-depleted cells are aromatic amino acid auxotrophs in *Mycobacterium smegmatis*”*

We believe this new title provides more information about our findings reported in this work.

11. All growth curves should be presented in log scale. P values for Fig 2C are warranted.

We have performed the changes indicated. Our Figure 5D (previous 2C) is now presented in log scale and we have provided p-values (on Figure 5D legend) for comparisons between control strains pNIP40::Ø or *aroA*_WT with strain *aroA*_D61W, showing that their growth differences are not statistically significant. As explained for Reviewer 2 (question 3.1), we decided to suppress the data from the growth curve of the reference strain *M. smegmatis* mc²155 because we realized this is not a suitable control to compare with *aroA*-disrupted strains. This strain have not passed by the same experimental procedures and do not have the vector pNIP40/b integrated.

12. I believe almost all of the supplemental material can be incorporated into the main text (and maybe some could be deleted if not essential to show).

As suggested, we have reorganized our figures and part of the data that was presented in the Supplemental Material was transferred to the main text. As supplemental information, we maintained the MESG synthesis description, the SDS-PAGE with the purification steps of WT *M*sEPSPS, mass spectra of point mutants produced and Table S1. We have also transferred our previous Figure 1 to Supplemental Material as indicated by Reviewer 2.

September 27, 2021

Prof. Cristiano Valim Bizarro
Pontifícia Universidade Católica do Rio Grande do Sul
Porto Alegre
Brazil

Re: Spectrum00009-21R1 (EPSP synthase-depleted cells are aromatic amino acid auxotrophs in *Mycobacterium smegmatis*)

Dear Prof. Cristiano Valim Bizarro:

Thank you for submitting your manuscript to Microbiology Spectrum. Your manuscript has been re-evaluated by all three reviewers, and all appreciated the significant improvement of this revision. However, there are still concerns about the appropriate statistics being performed. Figure 2 in particular requires the addition of statistical analysis. There are also suggestions for edits to other figures and text. It is likely these reviews can be adequately addressed through a text-only revision.

When submitting the revised version of your paper, please provide (1) point-by-point responses to the issues raised by the reviewers as file type "Response to Reviewers," not in your cover letter, and (2) a PDF file that indicates the changes from the original submission (by highlighting or underlining the changes) as file type "Marked Up Manuscript - For Review Only". Please use this link to submit your revised manuscript - we strongly recommend that you submit your paper within the next 60 days or reach out to me. Detailed information on submitting your revised paper are below.

Link Not Available

Sincerely,

Amanda Oglesby

Journals Department
Reviewer comments:

Reviewer #1 (Comments for the Author):

Most of the concerns were addressed.

However, one concern still remains. Authors pointed out that *M. smegmatis* and *M. tuberculosis* intracellular phenotypes are not the same and difficult to use *M. smegmatis* to represent intracellular *M. tuberculosis* phenotype. If this is true, this reviewer doesn't agree that *M. smegmatis* is a good model strain to validate the vulnerable metabolic steps of *M. tuberculosis* and propose as drug targets. Authors defended the critique raised by Reviewer 2 by mentioning that *M. smegmatis* is an optimal model to be experimentally feasible to represent virulent *M. tuberculosis* strain, which doesn't make sense to this reviewer. Many papers showed that intracellular *M. smegmatis* gradually loses its viability over time but have validated the gene essentiality by comparing intracellular CFU between WT and target mutants. This reviewer still thinks that intracellular essentiality of *aroA* mutant is an important piece of data to support author's claim.

In Fig 2, statistical analysis of strains before and after ATc treatment was missing.

Reviewer #2 (Comments for the Author):

In the revised manuscript by Dugue-Villeges, authors have made significant changes to the manuscript in response to previous review. The new manuscript better places the results in the context of the field and clarifies some results. There remain some addressable issues.

In the current manuscript the authors have constructed conditional mutants in *M. smegmatis* *aroA* using CRISPRi. The mutants have a growth defect in defined media lacking aromatic amino acid substitution. Growth curves in Figure 2 should have some statistical analyses to indicate whether differences observed at the 24h point are significant. This is more important for figure 2 than figure 5 since there are no differences in figure 5 growth, where p values are provided in legend. Curious if PAM2 actually gives significant growth difference especially in light of Figure 3 where Western analysis was used to demonstrate the reduced protein production upon knock down. First, these blots would benefit from better labeling. The upper blot is PAM1, 3 constructs. PAM2 is in panel B in duplicate. Duplicate is not necessary, but the quality of the blot for PAM2 could be improved. Densitometry could be performed to quantify protein levels.

Line 443-445, suggest: ...using the *mmpL3* gene as a positive control. This gene encodes for the essential MmpL3 protein that is responsible for TMM transport across the inner membrane for incorporation into new cell envelope.

Figure legend 1, line 895 I think this should reference Fig 1A.

Reviewer #3 (Comments for the Author):

This is a much improved manuscript and they authors addressed my concerns. However they did not indicate in what statistical tests they applied, even though they indicated statistics were performed.

Staff Comments:

Preparing Revision Guidelines

Please return the manuscript within 60 days; if you cannot complete the modification within this time period, please contact me. If you do not wish to modify the manuscript and prefer to submit it to another journal, please notify me of your decision immediately so that the manuscript may be formally withdrawn from consideration by Microbiology Spectrum.

Reviewer comments:

Reviewer #1 (Comments for the Author):

Most of the concerns were addressed.

1. However, one concern still remains. Authors pointed out that M. smegmatis and M. tuberculosis intracellular phenotypes are not the same and difficult to use M. smegmatis to represent intracellular M. tuberculosis phenotype. If this is true, this reviewer doesn't agree that M. smegmatis is a good model strain to validate the vulnerable metabolic steps of M. tuberculosis and propose as drug targets. Authors defended the critique raised by Reviewer 2 by mentioning that M. smegmatis is an optimal model to be experimentally feasible to represent virulent M. tuberculosis strain, which doesn't make sense to this reviewer. Many papers showed that intracellular M. smegmatis gradually loses its viability over time but have validated the gene essentiality by comparing intracellular CFU between WT and target mutants. This reviewer still thinks that intracellular essentiality of aroA mutant is an important piece of data to support author's claim.

As described in the main text of the revised version of our manuscript, by combining gene disruption, gene knockdown, point mutations and kinetic analysis of recombinant WT and mutant proteins, we provide robust data showing that, in *M. smegmatis*, (1) *aroA*-deficient cells are auxotrophic for AroAAs, (2) the growth impairment of *aroA*-knockdown cells can be rescued by providing AroAAs, and (3) catalytic residues of EPSPS are essential under growth conditions without AroAA supplementation, implying this enzyme activity in *AroA* gene essentiality. These results contrast with the current view that the mycobacterial shikimate pathway essentiality cannot be bypassed by the availability of aromatic compounds that are synthesized from pathways that start with chorismate, the end-product of the shikimate pathway. This view is mainly based on a knockout study on the SK-encoding *aroK* gene from *M. tuberculosis* (PMID: 12368440). We discussed our results in the context of studies performed in many other bacteria that indicate the general perception that the shikimate pathway is essential for bacteria, plants and fungi is rather oversimplistic. We focused in bringing this discussion into mycobacteria as well, by providing compelling and complementary data indicating this is not necessarily the case, and that drug development efforts towards EPSPS inhibition may be ineffective if bacilli have access to external sources of AroAAs in the context of infection.

As mentioned in our previous response to reviewer's 2 comments, we see as natural next steps for this work (1) to evaluate whether the same apply for *M. tuberculosis*, in other words, whether AroAA auxotrophy is also observed in *AroA*-deficient (or *AroA*-knockdown) *Mtb* cells, and, if this is the case, (2) evaluate whether the microenvironment experienced by infecting bacilli in a TB patient could provide aromatic compounds in sufficient amounts to render the shikimate pathway dispensable and efforts to develop drugs against enzymes of this pathway ineffective. The second goal, more difficult, can make use of different experimental systems, such as macrophage infection assays and animal models, aiming to mimic the microenvironment experienced by infecting bacilli during TB. In our view, the data provided in our study using *M. smegmatis* as a model system is sufficient and robust enough to justify this undertaking. It was not our purpose to tackle this second goal, which we think could be performed directly in *Mtb* cells.

2. In Fig 2, statistical analysis of strains before and after ATc treatment was missing.

As indicated, we have included the statistical analysis of the data provided. We detail the changes introduced into the main text to cope with the results of this analysis in the response to the first question of Reviewer #2.

Reviewer #2 (Comments for the Author):

In the revised manuscript by Dugue-Villeges, authors have made significant changes to the manuscript in response to previous review. The new manuscript better places the results in the context of the field and clarifies some results. There remain some addressable issues.

- 1. In the current manuscript the authors have constructed conditional mutants in *M. smegmatis* *aroA* using CRISPRi. The mutants have a growth defect in defined media lacking aromatic amino acid substitution. Growth curves in Figure 2 should have some statistical analyses to indicate whether differences observed at the 24h point are significant. This is more important for figure 2 than figure 5 since there are no differences in figure 5 growth, where p values are provided in legend. Curious if PAM2 actually gives significant growth difference especially in light of Figure 3 where Western analysis was used to demonstrate the reduced protein production upon knock down.**

As indicated, we have performed the statistical analyses of our data presented in Figure 2. We introduced the following piece of text in the Materials and Methods section (lines 283-285):

“Two-way ANOVA followed by Bonferroni’s multiple comparisons test was performed using GraphPad Prism version 7.0.0 for Windows, GraphPad Software, San Diego, California USA, www.graphpad.com.”

We found significant differences in growth at the 24h point for the three PAMs when cultures were grown in 7H9 medium. ($p < 0.01$ for PAM1 and PAM3 and $p < 0.05$ for PAM2 ($p = 0.0278$)). Interestingly, we also found a small but significant difference in growth for PAM 1 and 3 in LB medium ($p < 0.01$).

Therefore, we expanded our result section where these results are reported and described in more detail what we have found (lines 444 – 478):

*“We observed a similar growth pattern with or without ATc in liquid LB culture medium (Fig. 2B-D), although the levels of endogenous MsEPSPS were greatly reduced in *aroA*-knockdown cells with sgRNAs targeting sequences adjacent to any of the three motifs studied, as evaluated by immunoblot (Fig. 3A-B). By performing densitometric analysis, we found that the cellular levels of MsEPSPS in *aroA*-knockdown cells had a 5.7 (PAM1), 4.6 (PAM2) and 3.6 (PAM3) fold reduction, compared to the MsEPSPS levels of equivalent cultures grown without ATc. Interestingly, *aroA* knockdown directed to a sequence adjacent to the PAM sequence predicted to be of higher “strength” (20) (PAM1) resulted in the higher level of MsEPSPS reduction (5.7-fold), while the experiment targeting a sequence adjacent to the PAM sequence of predicted lower strength (PAM3) resulted in the lower level of MsEPSPS reduction observed (3.6-fold). In contrast to the similar growth curves observed in cultures grown on LB, we observed a marked decrease in bacterial growth at 24h in the presence of ATc, in cells grown in liquid (7H9) defined medium. This was observed for *aroA*-knockdown cells containing the CRISPRi system targeting sequences adjacent to any of the PAM motifs studied (PAM1, PAM2 and PAM3), indicating that *aroA* gene knockdown leads to a larger bacterial growth perturbation under these defined nutritional conditions (Fig. 2F-H). For the bacterial cells containing the CRISPRi targeting a sequence adjacent to PAM2 (Fig. 2C and 2G), there was no difference in growth for ATc+ and ATc- cultures after 24h of growth in LB medium (Fig. 2C, $p > 0.999$), while the cultures differed at this same time point when grown in 7H9*

medium (Fig. 2G, $p < 0.05$). On the other hand, for cells containing the CRISPRi system targeting sequences adjacent to PAM1 (Fig. 2B and 2F) and PAM3 (Fig. 2D and 2H), the growth difference between ATc-treated and untreated cultures after 24h was significant in both nutritional conditions ($p < 0.01$). However, the difference in growth was much smaller in LB cultures, when compared to 7H9 cultures (mean difference in OD of 0.17 (LB) and 0.97 (7H9) for PAM1 (Fig. 2B and 2F), and 0.29 (LB) and 0.78 (7H9) for PAM3 (Fig. 2D and 2H)). We observed the same pattern for cultures grown on solid media, with larger differences between ATc-treated and untreated cultures grown in defined medium (7H10) when compared to LB (Fig. 2B-H, insets). This dependency on medium composition for the growth perturbation found in *aroA*-knockdown cells is not observed in our control *mmpL3*-knockdown cells (Fig. 2A and 2E).”

In the piece of text above, we also included information related to the densitometric analysis of the immunoblot (issue 3), as detailed below.

We also adapted a small part of the Discussion section to cope with these findings (lines):

*“In contrast to what we found in defined medium (liquid 7H9 or solid 7H10), we observed similar growth patterns for *aroA*-knockdown cells grown on rich medium (liquid or solid LB) when compared to control cells (Fig. 2A-D). We found only a subtle change (although significant) in growth curves at 24h in rich medium for *aroA*-knockdown cells containing the CRISPRi system targeting sequences adjacent to PAM1 and PAM3 motifs (Fig. 2B and 2D). However, in both growth conditions (LB or 7H9/7H10), no viable *aroA*-deleted cells could be retrieved without AroAA supplementation. Our results support the notion that target essentiality and vulnerability are strongly dependent on nutritional context and care must be taken to extrapolate the information gathered from in vitro models to the context of the disease.”*

- 2. First, these blots would benefit from better labeling. The upper blot is PAM1, 3 constructs. PAM2 is in panel B in duplicate. Duplicate is not necessary, but the quality of the blot for PAM2 could be improved.**

We have prepared a new version of Figure 3 with better labeling. We hope it will be easier for the reader of our manuscript to capture the information provided by this figure. We also restricted the images to the parts containing the bands of interest, as a way to facilitate the interpretation of our results:

To cope with the changes in figure 3 labeling, we adapted the figure legend accordingly (Lines 928 – 952):

“Figure 3. aroA-knockdown cells are depleted of endogenous MsEPSPS. Immunoblot with anti-MtEPSPS polyclonal antibody of protein samples from *M. smegmatis* cells containing CRISPRi construct coding for sgRNA targeting sequence adjacent to PAM1, PAM2 or PAM3. *M. smegmatis* cells were grown on liquid LB medium. Lanes marked with (+) or (-) correspond to samples from cultures with or without induction with anhydrotetracycline (ATc), respectively. (A) Results for sgRNAs directed to PAM1 and PAM3 adjacent sequences. Lane M: ProSieve® Color Protein Markers (Lonza). Lanes corresponding to samples collected before induction with ATc (0h) are from cells with CRISPRi system targeting PAM1 (PAM1, 0h) or PAM3 (PAM3, 0h) adjacent sequence. Lanes corresponding to samples collected 18h after induction with (+) or without (-) ATc are also from cells with CRISPRi system targeting PAM1 (PAM1, 18h) or PAM3 (PAM3, 18h) adjacent sequence. MsEPSPS: purified MsEPSPS as positive control. (B) Results for sgRNA directed to PAM2 adjacent sequence. Lane M: ProSieve® Color Protein Markers (Lonza). Samples were collected before induction (PAM2, 0h) or 18h after induction (PAM2, 18h) with (+) or without (-) ATc. MsEPSPS: purified MsEPSPS as positive control.”

3. Densitometry could be performed to quantify protein levels.

As indicated, we performed the densitometric analysis of our immunoblot to quantify protein levels. To do that, we used the Image Lab version 6.1.0 Standard Edition software (Bio-Rad Laboratories, Inc) and followed the procedures recommended for relative quantification.

Figure R1: Densitometric analysis of Panel A

Figure R2: Densitometric analysis of Panel B

Based on this analysis, we observed the following reductions in MsEPSPS levels in *aroA*-knockdown cells containing the CRISPRi system targeting PAM1, PAM2 and PAM3 adjacent sequences:

PAM1 (NNAGCAT): 5.7-fold reduction

PAM2 (NNAGGAT): 4.6-fold reduction

PAM3 (NNAGCAG): 3.6-fold reduction

The densitometric analysis for PAM2 is an average of the duplicate experiments performed, which resulted in similar estimations (4.58 and 4.69-fold reduction). Interestingly, *aroA* knockdown directed to a sequence adjacent to the PAM sequence predicted to be of higher “strength” (PMID: 28165460), PAM1 sequence, resulted in the higher level of MsEPSPS reduction (5.7-fold), while the experiment

targeting the a sequence adjacent to the PAM sequence of predicted lower strength (PAM3) resulted in the lower level of MsEPSPS reduction (3.6-fold).

- 4. Line 443-445, suggest: ...using the mmpL3 gene as a positive control. This gene encodes for the essential MmpL3 protein that is responsible for TMM transport across the inner membrane for incorporation into new cell envelope.**

Indeed. We had already incorporated a piece of text explaining the function of MmpL protein in our previous version of the manuscript (lines 437 – 441):

“This gene codes for the mycobacterial membrane MmpL3 protein, which is responsible for trehalose monomycolate transport through the cell inner membrane (39). In M. tuberculosis and M. smegmatis, it was shown that silencing mmpL3 expression disrupts bacterial growth (20,40), leading to accumulation of TDM and cell death (41).”

- 5. Figure legend 1, line 895 I think this should reference Fig 1A.**

As indicated, we have included the reference to fig. 1A.

Reviewer #3 (Comments for the Author):

This is a much improved manuscript and the authors addressed my concerns. However they did not indicate in what statistical tests they applied, even though they indicated statistics were performed.

We have provided more information about the statistical analyses performed, as detailed in the responses to the issues raised by Reviewer #2.

November 17, 2021

Prof. Cristiano Valim Bizarro
Pontifícia Universidade Católica do Rio Grande do Sul
Porto Alegre
Brazil

Re: Spectrum00009-21R2 (EPSP synthase-depleted cells are aromatic amino acid auxotrophs in *Mycobacterium smegmatis*)

Dear Prof. Cristiano Valim Bizarro:

Your manuscript has been accepted, and I am forwarding it to the ASM Journals Department for publication. You will be notified when your proofs are ready to be viewed.

Sincerely,

Amanda Oglesby
Editor, Microbiology Spectrum
